# NewtonGen: Physics-Consistent and Controllable Text-to-Video Generation via Neural Newtonian Dynamics

**Yu Yuan**
Purdue University

**Xijun Wang**
Purdue University

**Tharindu Wickremasinghe**
Purdue University

**Zeeshan Nadir**
Samsung Research America

**Bole Ma**
Purdue University

**Stanley H. Chan**
Purdue University

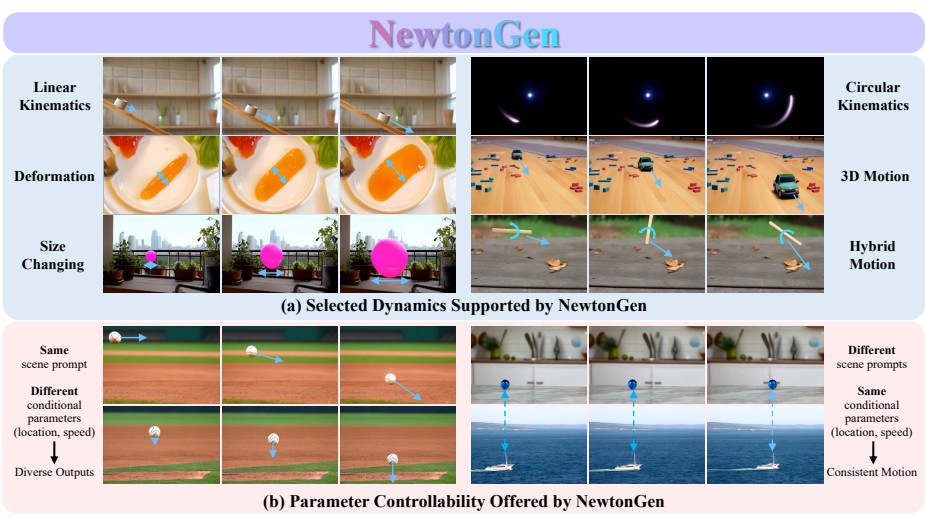

Figure 1: NewtonGen generates physically-consistent videos from text prompts, with diverse dynamic perception (a), and precise parameter control (b).

## Abstract

A primary bottleneck in large-scale text-to-video generation today is physical consistency and controllability. Despite recent advances, state-of-the-art models often produce unrealistic motions, such as objects falling upward, or abrupt changes in velocity and direction. Moreover, these models lack precise parameter control, struggling to generate physically consistent dynamics under different initial conditions. We argue that this fundamental limitation stems from current models learning motion distributions solely from appearance, while lacking an understanding of the underlying dynamics. In this work, we propose Newton-Gen, a framework that integrates data-driven synthesis with learnable physical principles. At its core lies trainable Neural Newtonian Dynamics (NND), which can model and predict a variety of Newtonian motions, thereby injecting latent dynamical constraints into the video generation process. By jointly leveraging data priors and dynamical guidance, NewtonGen enables physically consistent video synthesis with precise parameter control. All data and code are available at https://github.com/pandayuanyu/NewtonGen.

## 1 Introduction

Since the breakthrough of probabilistic diffusion models in the early 2020's (See, e.g., Ho et al. (2020); Song et al. (2021); Ramesh et al. (2021); Rombach et al. (2022)), foundational vision models

have created unprecedented opportunities for digital content generation. While contemporary video generators can synthesize visually appealing frames (Ho et al., 2022; OpenAI, 2024b; Hong et al., 2023; Peebles & Xie, 2023; Kong et al., 2024; Yang et al., 2025b), they struggle to produce dynamic sequences that adhere to physically plausible motion. For instance, many videos generated by these methods violate basic physical laws such as objects falling upward, or abruptly changing velocity and direction (Bansal et al., 2024; 2025; Zhang et al., 2025a; Li et al., 2025b; Duan et al., 2025; Motamed et al., 2025; Gu et al., 2025). The goal of this paper is to provide a solution to these issues.

The failures in the above situations, according to some literature, can potentially be remedied by scaling laws (Kaplan et al., 2020). However, recent researches such as Kang et al. (2025); Li et al. (2025a); Chefer et al. (2025); Lin et al. (2025); Bansal et al. (2024; 2025); Yuan et al. (2025a) consistently point to a deeper reason that current models only learn the distribution of visual appearances. They lack an understanding of the underlying physical laws. Existing frameworks typically treat videos as spatio-temporal tokens and optimize the likelihood at the pixel level. During inference, the models mainly rely on **memorization and imitation**, making it difficult to generalize to out-of-distribution scenarios (Kang et al., 2025). To bridge this gap, we argue that we need to explicitly incorporate physical laws into the learning process. This is not only a crucial step for video generation, but also essential for connecting generative AI with the physical world.

In this paper, we introduce **NewtonGen**, a novel framework that integrates a data-driven, pre-trained video generator with physics-informed, Neural Newtonian Dynamics (NND). In NND, we introduce a neural ordinary differential equation (neural ODE) model to learn and predict the Newtonian motion from physics-clean data. By learning the dynamics of motion and manipulating its initial physical states, we can predict physics-consistent trajectories, orientations, and shapes. Subsequently, a motion-controlled video generator produces diverse and realistic videos by conditioning on both the predicted states and scene prompts. In summary, the contribution of this paper is twofold:

1. We propose NewtonGen, a **physics-consistent and controllable** text-to-video framework that explicitly incorporates dynamics into the generation process, allowing for interpretable, white-box control over generated motion.

2. We introduce **Neural Newtonian Dynamics (NND)**, which models different dynamics via unified neural ordinary differential equations (ODEs). NND can efficiently learn the latent dynamics from a small amount of physics-clean data.

We conducted extensive experiments, showing that NewtonGen achieves physical consistency and controllability across various dynamics, as illustrated in Figure. 1, and outperforms other baselines.

## 2 RELATED WORK

### 2.1 VIDEO GENERATION MODELS

The emergence of diffusion models (Ho et al., 2020; Song et al., 2021) has greatly enhanced the ability of generative models to produce visually realistic images (Ramesh et al., 2021; Rombach et al., 2022; SD2). In video generation, models learn the distribution of real-world motion from large-scale datasets (OpenAI, 2024b; Blattmann et al., 2023; Hong et al., 2023; Yang et al., 2025b; Kong et al., 2024; NVIDIA, 2025; Wan et al., 2025). The method DiT proposed by Peebles & Xie (2023) introduces transformer architectures into diffusion models, further enhancing their scalability (Kaplan et al., 2020) for video generation tasks. Following this trend, representative video generation models (e.g., Sora OpenAI (2024b)) aim to leverage extensive video data to evolve toward general-purpose world simulators.

However, current video generation models still lack an understanding of real-world physics. Studies show that increasing data or model size does not help them learn the physical rules behind video content (Kang et al., 2025; Liu et al., 2025; Motamed et al., 2025; Lin et al., 2025). As a result, these models often produce videos that look realistic but contain physically incorrect dynamics when applied to out-of-distribution cases (Kang et al., 2025; Bansal et al., 2025; 2024; Meng et al., 2025; Zhang et al., 2025a; Li et al., 2025a; Gu et al., 2025; Chefer et al., 2025). The main reason is that they focus on appearance-level **motion** rather than the underlying **dynamics**.

## 2.2 PHYSICS-AWARE GENERATION

To address the challenge of physical plausibility in video generation, recent research efforts incorporate explicit physical priors into generative pipelines. Based on the stage and approaches of injecting physical knowledge, these methods can be broadly categorized into three types:

**Generation then Physical Simulation.** These methods first generate static 3D models or images conditioned on textual or visual inputs using generative models. Subsequently, physical simulation techniques such as Material Point Method (MPM) (Stomakhin et al., 2013) is applied to animate these static outputs into dynamic 3D scenes or videos (Lin et al., 2024; Xie et al., 2024; Tan et al., 2024; Zhang et al., 2024; Hsu et al., 2024). Although the post hoc physics-based rendering process is explicit and controllable, it demands significantly more manual effort. These methods can be summarized in the following Equation:

$$\widehat{\mathbf{V}} = \underbrace{P}_{\text{Physical Simulation}} \left( \overbrace{G_\psi(\mathbf{I})}^{\text{Video Generation}} \right) \tag{1}$$

where $P$ denotes the physical simulation, $G$ denotes the video generator parameterized by network weight $\psi$. $\mathbf{I}$ is the input conditional prompt or image, $\widehat{\mathbf{V}}$ is the video we want.

**Physical Simulation then Generation.** Approaches in this category (Yuan et al., 2023; Liu et al., 2024; Savant Aira et al., 2024; Chen et al., 2025; Xie et al., 2025; Li et al., 2025d) first apply physical simulation to conditionally specified images to generate plausible dynamic behaviors. The simulated dynamics are then utilized as conditional inputs for video generation models. For instance, PhysGen (Liu et al., 2024) segments dynamic objects from input images, simulates their motion according to Newtonian mechanics, and then refines the rendering by conditioning a video generation model on both simulated object positions and static backgrounds. However, generative models themselves lack any inherent physical reasoning or simulation capability: users must predefine the physical simulation parameters and rules for each scenario, and these settings cannot readily generalize to other contexts or different physical laws. These approaches can be summarized as:

$$\widehat{\mathbf{V}} = G_\psi\big(P(\mathbf{I})\big) \tag{2}$$

**Generation with Learned Physics Priors.** As illustrated in Equation 3, these methods leverage physical priors extracted from large-scale pretrained models to guide the generative process directly (Li et al., 2024; Lv et al., 2024; Xu et al., 2024; Yang et al., 2025a; Pandey et al., 2025; Xue et al., 2025; Cao et al., 2024; Wang et al., 2025; Yuan et al., 2025b; Zhang et al., 2025b; Chefer et al., 2025; Zhang et al., 2025c; Feng et al., 2025; Yang et al., 2025a). For example, PhyT2V (Xue et al., 2025) employs a large language model (LLM) ChatGPT (OpenAI, 2024a) and a vision–language model (VLM) (Wang et al., 2024a) as physics-consistency evaluators, performing multiple rounds of self-refinement to generate videos with improved physical plausibility. The main limitation of this line of work lies in the implicit assumption that existing models are capable of physical reasoning. In practice, however, these large-scale models, much like conventional video generation models, derive their so-called "physical understanding" purely from data fitting, and thus struggle when faced with physically challenging out-of-distribution scenarios. Our method broadly fits within this paradigm; however, our physical prior is driven by both explicit physics models and physics-clean data, which gives it stronger conditional controllability and better out-of-distribution generalization.

$$\widehat{\mathbf{V}} = G_\psi\big(P_\phi(\mathbf{I})\big) \tag{3}$$

where $\phi$ is the learned physical parameters.

## 2.3 LEARN PHYSICS FROM VIDEOS

Leveraging the spatiotemporal information in videos, methods such as Wu et al. (2015); Watters et al. (2017); Wu et al. (2017); Belbute-Peres et al. (2018); Raissi et al. (2019); Chari et al. (2019); Greydanus et al. (2019); Lutter et al. (2019); Zhong & Leonard (2020); Jaques et al. (2020); Le Guen & Thome (2020); Hofherr et al. (2023); Garrido et al. (2025); Garcia et al. (2025); Deng et al. (2025); Li et al. (2025c) estimate the parameters of known governing equation, which in turn enables tasks

such as future-frame prediction and physical reasoning. These methods usually adopt an encoder-decoder structure. Each frame is encoded into a latent physical state using models like a variational autoencoder (VAE) (Kingma & Welling, 2013; 2019). The latent state is then processed by a physics engine and decoded back to reconstruct the frame for training. Most of these approaches are designed for a single type of simple dynamical system. They are difficult to generalize to different systems within a single framework.

Our Neural Newtonian Dynamics (NND) is partly inspired by the aforementioned methods. We adopt an encoder-only architecture integrated with physics-informed general neural ordinary differential equations (ODEs) to explicitly capture diverse dynamics from videos.

## 3 PRELIMINARY CONCEPTS

### 3.1 INCORPORATING PHYSICAL DYNAMICS INTO DATA-DRIVEN VIDEO GENERATION

Existing video generation models (OpenAI, 2024b; Hong et al., 2023; Kong et al., 2024; Yang et al., 2025b; Blattmann et al., 2023) are mostly data-driven, relying on large-scale video datasets without physical annotations. While they achieve good performance within training domains, they often fail in out-of-distribution scenarios by violating basic physical laws (Chefer et al., 2025; Kang et al., 2025). In contrast, physics-driven dynamics methods explicitly incorporate governing constraints, yielding better physical plausibility and out-of-distribution generalization (Champion et al., 2019). To combine the strengths of both, we propose incorporating physical dynamics into data-driven video generation. This hybrid paradigm leverages the low-bias learning capacity of data-driven models while injecting lightweight dynamics priors to enforce consistency with fundamental laws, thereby achieving improved generalization and physically coherent video synthesis.

### 3.2 MODELING THE DYNAMICS IN A GENERAL PHYSICS-INFORMED NEURAL ODE

To understand how NewtonGen works, we first ask: what is the best way to describe Newtonian motion? Physics textbooks tell us that if we are given the initial position, initial velocity, acceleration and mass, we can predict the trajectory of how the object moves in space and time. In mathematics, this is done through ordinary differential equations (ODEs). Based on this intuition, we consider a second-order system governed by autonomous ODEs with no explicit time-varying external forces. We constrain the ODEs to the second order, because most common physical motions in daily life (e.g., flying balls) can generally be described by second-order dynamics. Even in more complex motions and three-dimensional scenes, the dynamics can still be effectively characterized by second-order formulations over relatively short time intervals with sufficiently dense anchor points.

To handle a wide range of video generation tasks, we require the ODE framework capable of accommodating diverse dynamics. This raises the following question: how can we construct a universal ODE framework that can describe various types of motion? To this end, we introduce two key design principles:

1. **Latent Physical States.** We define a 9-dimensional latent physical state vector $\mathbf{Z} = [x, y, v_x, v_y, \theta, \omega, s, l, a]$. Here, $x, y$ represent the position, and $v_x, v_y$ represent velocity of the object's center of mass. $\theta, \omega$ encode the object's rotation or rotation about a pivot point. $s, l$ are the object's shortest and longest dimensions, and $a$ is its projected area. This formulation allows our physical states to capture translation, rotation, deformation, and other complex behaviors. 3D motion effect can also be equivalently realized through the combination of position and size control.

2. **Linear Physics-Informed Neural ODEs with a Residual MLP.** Different motions follow inherently different dynamical laws: for instance, free-fall can be described by a simple linear ODE, while a damped pendulum or other unknown motion cannot. To address this, we combine linear physics-informed neural ODEs with a residual multilayer perceptron (MLP) as illustrated in Equation 4 and Figure. 2(a). The linear ODEs capture the dominant linear dynamics, while the residual MLP models nonlinear and unknown components, enabling the system to flexibly approximate a wide range of physical behaviors.

$$a_z \, \ddot{z} + b_z \, \dot{z} + c_z \, z + d_z + \text{MLP}(\mathbf{Z}) = 0 \tag{4}$$

where $z$ is one element of the 9-dimensional latent physical state vector $\mathbf{Z}$, and $a_z, b_z, c_z, d_z$ are learnable parameters of the linear ODE. We can use multiple ODEs to predict future physical states in a compact autonomous form:

$$\mathbf{Z}_t = \mathbf{Z}_0 + \int_{t_0}^{t} \text{Func}\big(\mathbf{Z}(\tau)\big) \, \mathrm{d}\tau, \tag{5}$$

where $\text{Func}\big(\mathbf{Z}(\tau)\big)$ represents the collection of all individual $\mathrm{d}z/\mathrm{d}t$ ODEs, and $\mathbf{Z}_0 = \mathbf{Z}(t_0)$ is the known initial physical state at time $t_0$.

## 4 METHODOLOGY

**Problem Definition.** We study generating videos of foreground objects whose motion obey classical mechanics, with controllable physical parameters, from prompts describing the scene and initial conditions.

**Overall Framework.** As shown in Figure. 2, NewtonGen consists of two main stages. As illustrated in Figure. 2(b), in the first stage, we train the proposed Neural Newtonian Dynamics (NND) on a small set of physics-clean data to learn the underlying motion dynamics and parameters. In the second stage shown in Figure. 2(c), we use the learned dynamics to predict future physical states from arbitrary initial conditions specified by the user via text prompts at inference, and feed these predictions, together with the scene prompt, into a motion-controlled text-to-video generation model to produce the final video.

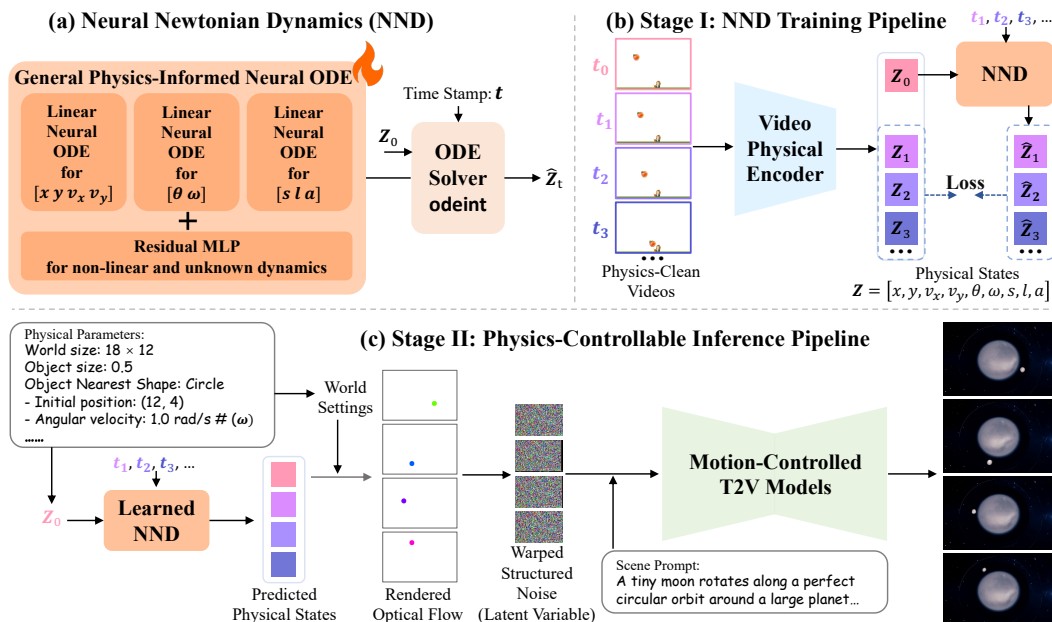

Figure 2: The overall framework of NewtonGen. a) Neural Newtonian Dynamics (NND) employs physics-informed linear neural ODEs combined with an MLP to build a general dynamics learning framework suitable for diverse motions. b) We train NND on a physics-clean dataset to capture the underlying dynamics. c) Using the learned NND together with a data-driven motion-controlled model, we generate physically plausible and controllable videos.

### 4.1 NEURAL NEWTONIAN DYNAMICS

As discussed in Subsection 3.2, our Neural Newtonian Dynamics (NND) aims to construct a unified model capable of capturing a wide range of dynamical behaviors. Its core is composed of physics-informed general neural ordinary differential equations (Neural ODEs) (Chen et al., 2018). As

demonstrated in Figure. 2(a), physics-informed linear neural ODEs are employed to model the underlying linear dynamics, while a residual three-layer MLP captures nonlinear and unknown components of the dynamics. With this design, the learnable neural ODEs can represent more complex or real-world dynamics. Given an initial physical states $\mathbf{Z}_0$ and a time stamp $t$, the ODE solver `odeint` (Chen et al., 2018) can be used to predict the object's future physical states $\mathbf{Z}_t$.

## 4.2 TRAINING FOR NEURAL NEWTONIAN DYNAMICS

**Overall Training Pipeline.** Figure. 2(b) illustrates that, for training Neural Newtonian Dynamics (NND), we adopt an encoder-only architecture. This design does not require decoding back to images, and optimizes solely in the latent physical space, significantly reducing computational cost. Specifically, a Video Physical Encoder $E_{phys}$ compresses each video frame into its corresponding physical state. The initial state $Z_0$ and the sequence of frame time stamps $(t_1, t_2, t_3, \dots)$ are fed into NND, which predicts $(\widehat{\mathbf{Z}}_1, \widehat{\mathbf{Z}}_2, \widehat{\mathbf{Z}}_3, \dots)$. The loss is then computed between the predicted states and the states $(\mathbf{Z}_1, \mathbf{Z}_2, \mathbf{Z}_3, \dots)$ extracted by the Encoder $E_{phys}$ :

$$\text{Loss} = \frac{1}{T} \sum_{t=1}^{T} \| \underbrace{E_{phys}(\mathbf{I}_t)}_{\mathbf{Z}_t} - \underbrace{\text{NND}_\kappa(E_{phys}(\mathbf{I}_0), t)}_{\widehat{\mathbf{z}}_t} \|_2^2 \tag{6}$$

where $T$ denotes the number of sampled time stamps, $\mathbf{I}_t$ is the video frame at time $t$, and $\kappa$ represents the learnable parameters of the ODEs.

**Training Data.** To enable Neural Newtonian Dynamics to learn accurate and effective representations of physical dynamics, we require "physics-clean" video data. That is, the motion in the videos should be prominent and monotonic, with no motion blur or excessive noise in each frame, and minimal color, texture, or background distractions. However, to our knowledge, such high-quality datasets of physical dynamics are still lacking. To address this, we developed a Python-based physics data simulator that can render videos with precise timestamps for different world settings, initial conditions, and types of dynamics. More details are shown in the Supplementary Material.

**Video Physical Encoder.** To extract physical state labels from videos, we first apply the visual segmentation foundation model SAM2 (Ravi et al., 2025) to obtain masks for the dynamic regions in each frame. From the extracted masks, we extract the centroid, area, long/short axes, and orientation of the foreground mask via morphological analysis, and compute velocities from inter-frame differences. Finally, these attributes are uniformly quantized to form the physical states $\mathbf{Z}$.

## 4.3 INFERENCE FOR PHYSICAL-CONTROLLABLE TEXT-TO-VIDEO GENERATION

As illustrated in Figure. 2(c), during inference, we decouple physical dynamics reasoning from video generation. Physical dynamics reasoning focuses on modeling and predicting the motion of dynamic objects, while video generation leverages rich scene understanding and generation capabilities to render detailed and flexible visual content.

We adopt Go-with-the-Flow (Burgert et al., 2025) as our base video generation model, which achieves motion control through structured noise (Chang et al., 2024). By warping the independently initialized Gaussian noise of each frame according to the input optical flow, temporal correlations emerge between the initial noise of consecutive frames, leading to more effective motion control. Other motion-controlled video generation models (Yin et al., 2023; Wang et al., 2024b; Zhang et al., 2025d) typically encode trajectories or bounding boxes through ControlNet (Zhang et al., 2023) or additional encoders and inject the features into the base video generators. However, these approaches often struggle with handling deformations, rotations, or more complex motions. We choose Go-with-the-Flow for its generality and effectiveness.

To effectively transfer the physical knowledge from our NND to the video generation model, a multi-step procedure is required. First, based on the user's physical prompts, we parse the initial physical state $\mathbf{Z}_0$ and future time stamps. $\mathbf{Z}_0$ and the frame timestamps are fed into the trained NND to obtain the corresponding physical states for all future frames. Next, using the world setting information parsed from the physical prompts (e.g., scene dimensions, object size, and the closest simple geometric shape of the object), we compute an approximate pixel-level optical flow for each frame based on the predicted physical states. These flows are then temporally and spatially

downsampled to match the resolution of the video generator's latent space, resulting in a structured optical flow sequence. Finally, combining the user's scene prompts, video frames are sampled to produce the final videos.

## 5 EXPERIMENTS

In this section, we evaluate the applications of our framework for physically-consistent and controllable video generation. Subsection 5.1 presents implementation details, Subsection 5.2 compares NewtonGen with other baselines, and Subsection 5.3 discusses the results of ablation study.

### 5.1 IMPLEMENTATION DETAILS

**Supported Motion Types.** In NewtonGen, we evaluate 12 distinct types of motion: **uniform motion, acceleration, deceleration, parabolic motion, 3D motion, slope sliding, circular motion, rotation, parabolic motion with rotation, damped oscillation, size changing, and deformation**. These categories cover the most common fundamental motion patterns encountered in everyday scenarios. The tested velocity magnitudes are mostly within the range of 0–15 m/s, while the duration of the generated motions is typically concentrated within 1–2 seconds.

**Training Details for NND.** We optimize the NND with the AdamW optimizer (initial learning rate $1 \times 10^{-4}$) and a CosineAnnealingLR scheduler (Loshchilov & Hutter, 2017). For each type of motion, we collect 100 physical videos with different initial conditions from the physics simulator mentioned in Subsection 4.2 as training data. The model is trained with a batch size of 64 for a total of 20,000 epochs, which requires about 2 hours on a single NVIDIA A100 80 GB GPU.

**Metrics.** Assessing the physical consistency of different video generation models is hampered by the absence of a shared ground truth and by the fact that each synthetic sequence is defined in its own coordinate frame and scale and time. Consequently, a single, unified physical evaluation metric cannot be directly applied. In this work, we are inspired by the Physical Invariance Score (PIS) (Zhang et al., 2025a), which evaluates physical plausibility by checking whether a motion preserves its expected invariants $C$. For example, in parabolic motion and the horizontal velocity $v_x$ should be constant. We use SAM2 (Ravi et al., 2025) to segment the object in each frame and obtain its centroid and shape features. Velocities are estimated from frame-to-frame centroid differences. The Physical Invariance Score for a quantity $C$ is defined as the relative standard deviation of $C$ over time:

$$\text{PIS} = (1 + C_\sigma/(|C_\mu| + \epsilon))^{-1} \tag{7}$$

where $C$ denotes one of the quantities introduced above (i.e., the horizontal velocity, the vertical acceleration, or the angular speed). $\epsilon = 1 \times 10^{-5}$ is added to the denominator to prevent division by zero. The PIS score is bounded in $[0, 1]$, with a value of 1 indicating that the evaluated physical quantity remains perfectly invariant.

We also adopt the background consistency (BC) and motion smoothness (MS) metrics from VBench ((Huang et al., 2024)) to assess scene consistency and motion quality.

### 5.2 COMPARISONS WITH OTHER METHODS

**General Comparisons.** We compare our method with five baselines: SORA (OpenAI, 2024b), Veo3 (Google, 2025), CogVideoX-5B (Yang et al., 2025b), Wan2.2 (Wan et al., 2025) and PhyT2V (Xue et al., 2025). These baselines represent the current state-of-the-art in both closed-source and open-source video generation models, as well as physics-based generation methods. We standardize the video generation settings across all methods to ensure maximum fairness in comparison. We collected 24 prompts for each motion type to assess the physical generation capabilities of all methods.

In Figure. 3, the video sequences generated by NewtonGen exhibit the highest degree of physical consistency across all 12 motion types. The motions display smooth and realistic trajectories without abrupt changes in direction or speed, realistic 3D movement effects (with object scale gradually increasing as distance decreases), physically plausible self-rotation (objects preserve shape with uniform angular velocity), smooth deformations (edges stretch or shrink progressively), and natural size variations (e.g., balloon diameter increases over time but at a decelerating rate). Table 1 further shows that our model achieves significantly higher physical consistency scores than competing

methods across different motion categories. Notably, some motion types do not admit perfect physical invariants; in these cases, we still compute quantities such as angular velocity and compare them against reference simulation videos under the same conditions.

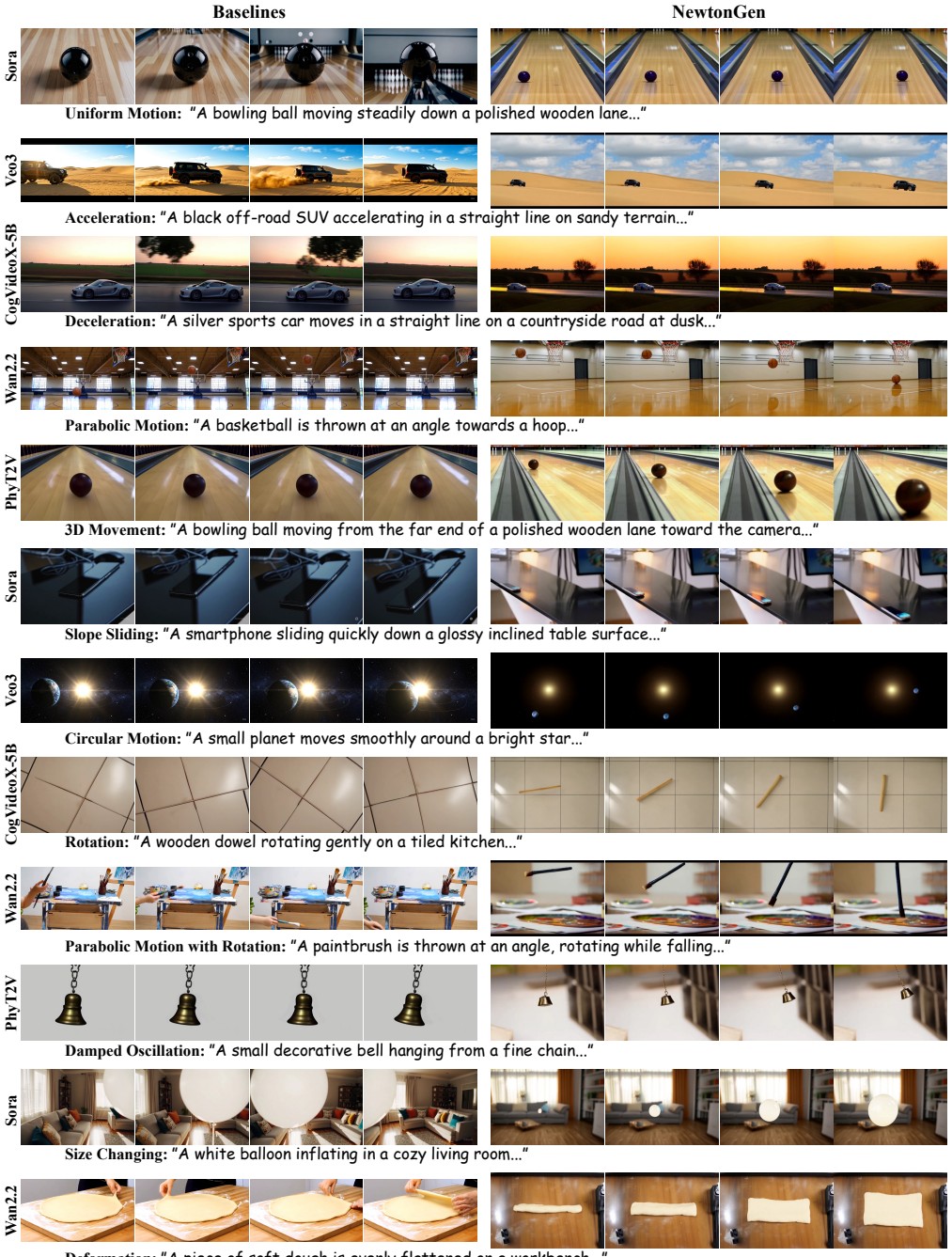

Figure 3: Visual comparisons of different text-to-video generation methods across diverse physical dynamics, where our method consistently shows strong physical consistency and controllability (such as we can control the shape of dough).

**Parameter Controllability Comparisons.** In Figure. 4, we demonstrate NewtonGen's ability to perceive physical parameters. Unlike other models, our method faithfully reflects world settings,

Table 1: Quantitative comparison with different methods. The reference scores are computed on the simulated videos, and the detailed definition of PIS for each type of motion is provided in the Appendix. BC and MS denote background consistency and motion smoothness, respectively, with values in parentheses indicating the standard deviation across videos. We highlight the best and second-best results for each metric.

| Motion Types | Metrics↑ | Reference | Sora | Veo3 | CogVideoX-5B | Wan2.2 | PhyT2V | Ours |
|---|---|---|---|---|---|---|---|---|
| | | | | | Methods | | | |
| Uniform Motion | PIS-$v$ | 0.9972 | 0.6548(0.022) | 0.9784(0.006) | 0.5392(0.007) | 0.6395(0.029) | 0.5349(0.014) | 0.9830(0.005) |
| | BC | 1.0000 | 0.9573(0.003) | 0.9491(0.024) | 0.9534(0.018) | 0.9683(0.027) | 0.9612(0.015) | 0.9694(0.020) |
| | MS | 1.0000 | 0.9926(0.003) | 0.9953(0.001) | 0.9905(0.005) | 0.9939(0.003) | 0.9876(0.015) | 0.9962(0.003) |
| Acceleration (Uniform) | PIS-$a_x$ | 0.8489 | 0.3437(0.355) | 0.6187(0.308) | 0.5458(0.038) | 0.3077(0.261) | 0.5033(0.011) | 0.6568(0.013) |
| | BC | 1.0000 | 0.9495(0.011) | 0.9373(0.015) | 0.9518(0.037) | 0.9695(0.018) | 0.9636(0.021) | 0.9748(0.012) |
| | MS | 1.0000 | 0.9852(0.011) | 0.9909(0.004) | 0.9876(0.008) | 0.9908(0.005) | 0.9822(0.010) | 0.9918(0.009) |
| Deceleration (Uniform) | PIS-$a_x$ | 0.8872 | 0.6162(0.072) | 0.6173(0.102) | 0.4988(0.014) | 0.4705(0.328) | 0.5167(0.023) | 0.6891(0.007) |
| | BC | 1.0000 | 0.9494(0.026) | 0.9295(0.039) | 0.9623(0.017) | 0.9721(0.012) | 0.9622(0.012) | 0.9744(0.012) |
| | MS | 1.0000 | 0.9883(0.006) | 0.9933(0.003) | 0.9787(0.024) | 0.9903(0.007) | 0.9814(0.014) | 0.9947(0.005) |
| Parabolic Motion | PIS-$v_x$ | 0.9988 | 0.9095(0.014) | 0.9042(0.012) | 0.7392(0.007) | 0.7747(0.126) | 0.6370(0.199) | 0.9803(0.002) |
| | PIS-$a_y$ | 0.9487 | 0.5723(0.266) | 0.7662(0.139) | 0.4230(0.028) | 0.5571(0.953) | 0.3567(0.799) | 0.8189(0.014) |
| | BC | 1.0000 | 0.9486(0.023) | 0.9514(0.023) | 0.9330(0.030) | 0.9602(0.028) | 0.9436(0.046) | 0.9693(0.014) |
| | MS | 1.0000 | 0.9915(0.004) | 0.9948(0.002) | 0.9856(0.009) | 0.9903(0.007) | 0.9844(0.011) | 0.9967(0.001) |
| 3D Motion | PIS-$\Delta_l$ | 0.7388 | 0.5013(0.005) | 0.5932(0.005) | 0.3026(0.005) | 0.4583(0.005) | 0.2911(0.007) | 0.6472(0.005) |
| | PIS-$v_y$ | 0.9986 | 0.8481(0.008) | 0.8913(0.008) | 0.6690(0.003) | 0.8384(0.018) | 0.6510(0.002) | 0.9371(0.007) |
| | BC | 1.0000 | 0.9426(0.017) | 0.9410(0.022) | 0.9620(0.018) | 0.9772(0.008) | 0.9629(0.016) | 0.9672(0.018) |
| | MS | 1.0000 | 0.9934(0.003) | 0.9944(0.003) | 0.9945(0.003) | 0.9943(0.002) | 0.9888(0.012) | 0.9954(0.005) |
| Slope Sliding | PIS-$a_x$ | 0.8741 | 0.4931(0.153) | 0.6081(0.157) | 0.3533(0.160) | 0.3108(0.421) | 0.3570(0.354) | 0.6312(0.041) |
| | PIS-$a_y$ | 0.9148 | 0.4616(0.212) | 0.3815(0.092) | 0.4731(0.028) | 0.3967(0.744) | 0.4297(0.569) | 0.5840(0.043) |
| | BC | 1.0000 | 0.9667(0.013) | 0.9631(0.016) | 0.9556(0.024) | 0.9653(0.017) | 0.9568(0.022) | 0.9787(0.010) |
| | MS | 1.0000 | 0.9919(0.005) | 0.9958(0.002) | 0.9903(0.006) | 0.9912(0.005) | 0.9829(0.014) | 0.9971(0.001) |
| Circular Motion | PIS-$\omega$ | 0.9933 | 0.8393(0.010) | 0.8932(0.007) | 0.7726(0.026) | 0.4677(0.006)) | 0.6391(0.322) | 0.9788(0.018) |
| | BC | 1.0000 | 0.9684(0.012) | 0.9711(0.010) | 0.9842(0.013) | 0.9745(0.016) | 0.9677(0.027) | 0.9812(0.007) |
| | MS | 1.0000 | 0.9949(0.001) | 0.9960(0.001) | 0.9979(0.001) | 0.9949(0.004) | 0.9974(0.002) | 0.9980(0.001) |
| Rotation (Uniform) | PIS-$\omega$ | 0.9836 | 0.4267(0.099) | 0.5285(0.436) | 0.6596(0.023) | 0.3425(0.172) | 0.7842(0.304) | 0.8838(0.038) |
| | BC | 1.0000 | 0.9543(0.030) | 0.9650(0.018) | 0.9397(0.025) | 0.9620(0.010) | 0.9375(0.028) | 0.9700(0.008) |
| | MS | 1.0000 | 0.9900(0.006) | 0.9942(0.003) | 0.9795(0.028) | 0.9909(0.007) | 0.9878(0.006) | 0.9958(0.002) |
| Para. w/ Rotation | PIS-$v_x$ | 0.9990 | 0.5797(0.150) | 0.7029(0.197) | 0.6488(0.031) | 0.6558(0.175) | 0.7689(0.039) | 0.9446(0.008) |
| | PIS-$a_y$ | 0.9657 | 0.4903(2.581) | 0.5603(1.012) | 0.2614(0.127) | 0.4331(1.982) | 0.2879(0.046) | 0.5614(0.028) |
| | PIS-$\omega$ | 0.9829 | 0.6522(0.556) | 0.9019(0.119) | 0.3380(0.199) | 0.3474(0.334) | 0.4119(0.105) | 0.9289(0.029) |
| | BC | 1.0000 | 0.9532(0.016) | 0.9583(0.018) | 0.9567(0.018) | 0.9617(0.027) | 0.9675(0.020) | 0.9786(0.009) |
| | MS | 1.0000 | 0.9889(0.005) | 0.9952(0.001) | 0.9841(0.018) | 0.9908(0.005) | 0.9921(0.003) | 0.9969(0.001) |
| Damped Oscillation | PIS-$a_y$ | 0.9402 | 0.4418(0.364) | 0.3516(0.482) | 0.3083(0.055) | 0.3494(0.395) | 0.2841(0.042) | 0.5240(0.017) |
| | BC | 1.0000 | 0.9738(0.013) | 0.9666(0.011) | 0.9699(0.007) | 0.9715(0.014) | 0.9708(0.010) | 0.9743(0.012) |
| | MS | 1.0000 | 0.9909(0.014) | 0.9958(0.001) | 0.9853(0.005) | 0.9919(0.009) | 0.9867(0.003) | 0.9968(0.001) |
| Size Changing | PIS-$\Delta_r$ | 0.8501 | 0.2840(0.010) | 0.4167(0.006) | 0.5774(0.007) | 0.1972(0.022) | 0.4010(0.011) | 0.6362(0.010) |
| | BC | 1.0000 | 0.9507(0.025) | 0.9548(0.017) | 0.9636(0.019) | 0.9735(0.018) | 0.9666(0.022) | 0.9669(0.015) |
| | MS | 1.0000 | 0.9916(0.003) | 0.9955(0.002) | 0.9926(0.007) | 0.9889(0.008) | 0.9925(0.004) | 0.9955(0.002) |
| Deformation | PIS-$\Delta_l$ | 0.9247 | 0.3626(0.004) | 0.3466(0.017) | 0.3550(0.002) | 0.3515(0.043) | 0.3601(0.003) | 0.5492(0.005) |
| | BC | 1.0000 | 0.9553(0.039) | 0.9058(0.052) | 0.9462(0.018) | 0.9347(0.042) | 0.9211(0.010) | 0.9475(0.025) |
| | MS | 1.0000 | 0.9941(0.004) | 0.9940(0.006) | 0.9935(0.009) | 0.9903(0.009) | 0.9867(0.001) | 0.9957(0.001) |

Table 2: Quantitative results of ablation study. We compute the normalized absolute error between the predicted and ground-truth physical states across all time steps within the test batch.

| Motions | Uni | Acc | Dec | Para | 3DMot | Slope | Circ | Rota | ParaRota | Osci | Size | Def |
|---|---|---|---|---|---|---|---|---|---|---|---|---|
| Ablations | | | | | | Normalized Absolute Error ↓ | | | | | | |
| W/o MLP | 0.0174 | 0.0069 | 0.0104 | 0.0193 | 0.0937 | 0.0831 | 0.5388 | 0.0382 | 0.7451 | 0.2275 | 0.1239 | 0.0854 |
| Our-data10 | 0.0632 | 0.0260 | 0.0184 | 0.0284 | 0.1079 | 0.0935 | 0.1246 | 0.0739 | 0.0273 | 0.1045 | 0.2327 | 0.0555 |
| Our-data100 | 0.0142 | 0.0034 | 0.0078 | 0.0042 | 0.0182 | 0.0324 | 0.0255 | 0.0058 | 0.0064 | 0.0425 | 0.1193 | 0.0357 |
| Our-data500 | 0.0195 | 0.0051 | 0.0072 | 0.0040 | 0.0192 | 0.0307 | 0.0196 | 0.0049 | 0.0063 | 0.0694 | 0.1379 | 0.0290 |

object properties, and initial conditions, with trajectories and velocities that better follow physical laws (third row). More cases are provided in the Appendix.

## 5.3 ABLATION STUDY

Our ablation study focuses on the effects of the MLP in Neural Newtonian Dynamics (NND), the training data scale and real-world video training. As shown in Table 2, adding the MLP significantly improves NND's performance on nonlinear dynamics and noisy data. Increasing the training dataset

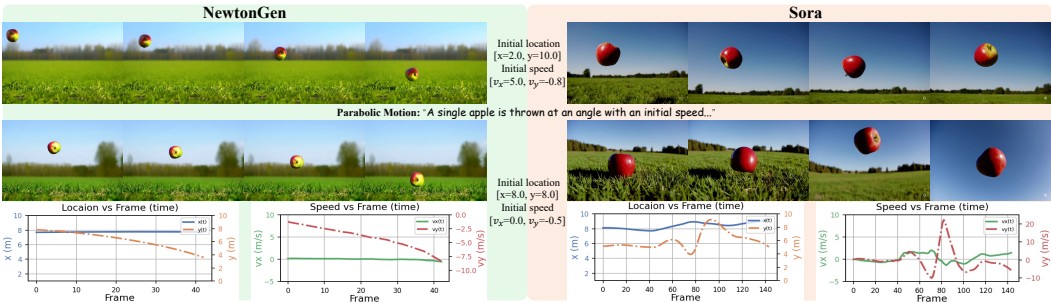

Figure 4: NewtonGen generates videos that can accurately reflect user-specified initial physical parameters, including object position, velocity, angle, shape and size.

size does not lead to notable gains, indicating that NND can accurately infer the underlying system dynamics from a relatively small number of physically clean samples.

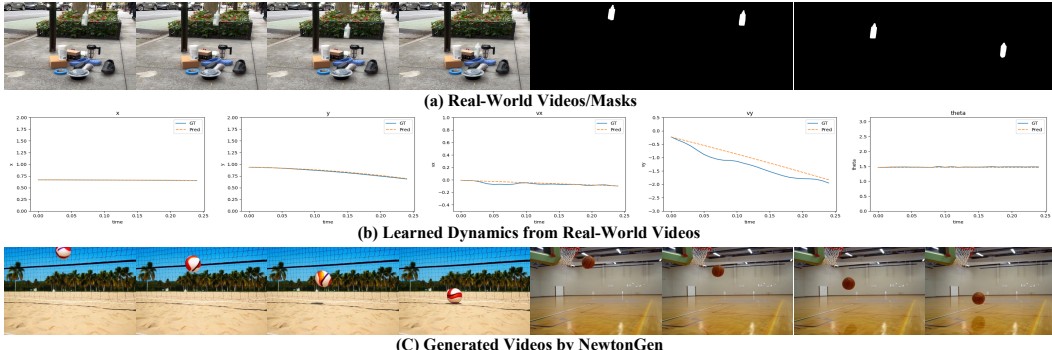

Figure 5: Feasibility of training NewtonGen(NND) on real-world videos

**Real-world video training.** To evaluate NewtonGen on real-world data, we select real-world videos from the PISABench (Li et al., 2025a) (example frames are shown in Fig. 5(a)). We train NND on these fall videos. As shown in Fig. 5(b), NND can effectively learn the underlying falling dynamics, even though the videos contain motion blur. Using the dynamics learned from these real videos, NewtonGen can then generalize to generate physically plausible falling motions in new scenes, as illustrated in Fig. 5(c). For this real-world case, the PIS-$v_x$ and PIS-$a_y$ scores are 0.8485 and 0.6008, which are lower than our results on the simulated dataset (0.9803 and 0.8189). This shows that collecting real dynamic scenes is feasible but time-consuming and requires careful setup (e.g., arranging scenes and measuring physical scales), and the data quality is often lower than that of clean simulations. Our simulated physics-clean data provides a faster and cheaper way to obtain high-quality training data.

## 6 CONCLUSION

In this paper, we introduce NewtonGen, a physics-consistent and controllable text-to-video generation framework. NewtonGen integrates a Neural Newtonian Dynamics (NND) module, which learns latent dynamics for diverse motions from a small set of physically accurate examples and predicts future physical states. We validate NewtonGen on over twelve different dynamic video generation tasks, demonstrating its physical consistency and parameter controllability. NewtonGen holds the potential to narrow the gap between current generative models and the real physical world.

**Limitations.** While our framework effectively models and predicts the dynamics of most common motions, it is based on continuous dynamics. This means that NewtonGen can be less effective for handling multi-object interactions (e.g., collisions or coalescence). We expect that future work incorporating event-based or discrete neural architectures will address these limitations.

# 7 ETHICS STATEMENT AND REPRODUCIBILITY STATEMENT

**Ethics Statement.** This model is designed to generate high-quality content and educational videos; however, when misused without labels and watermarks, it can produce fake videos and lead to the spread of misinformation.

**Reproducibility Statement.** All data, code and model weights will be made publicly available. In our model evaluation, we fix the random seed and provide the test prompts and generated videos in the supplementary materials and appendix. In addition, we include more detailed explanations of the evaluation metrics in the Appendix.

ACKNOWLEDGMENTS

This work is supported, in part, by the United States National Science Foundation under the grants 2133032, 2431505, and a research award from Samsung Research America.

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

# Appendix

## A  APPENDIX INTRODUCTION

This appendix provides additional discussions and details on the physics-clean video simulator (Section B), Neural Newtonian Dynamics network design and prediction accuracy analysis (Section C), evaluation details (Section D), more visual results (Section E), and a Q & A section (Section F). To illustrate the continuity and effects of physical coherence and controllability, **we recommend that readers view the Videos** included in the Project Page https://yuyuanspace.com/NewtonGen/.

## B  MORE DETAILS OF THE DATA SIMULATOR

For each type of motion, we construct a physics-clean dataset for training the NND model. Our simulator is built upon physical principles and renders videos with time stamps. The simulator supports multi-parameter control, including initial position, velocity, orientation, angular velocity, world settings (world size, friction coefficient, acceleration/deceleration coefficient, damping coefficient, pivot point), and object properties (size, shape). Representative samples are shown in Figure. 6, while the complete simulator code and additional video examples are provided at https://github.com/pandayuanyu/NewtonGen.

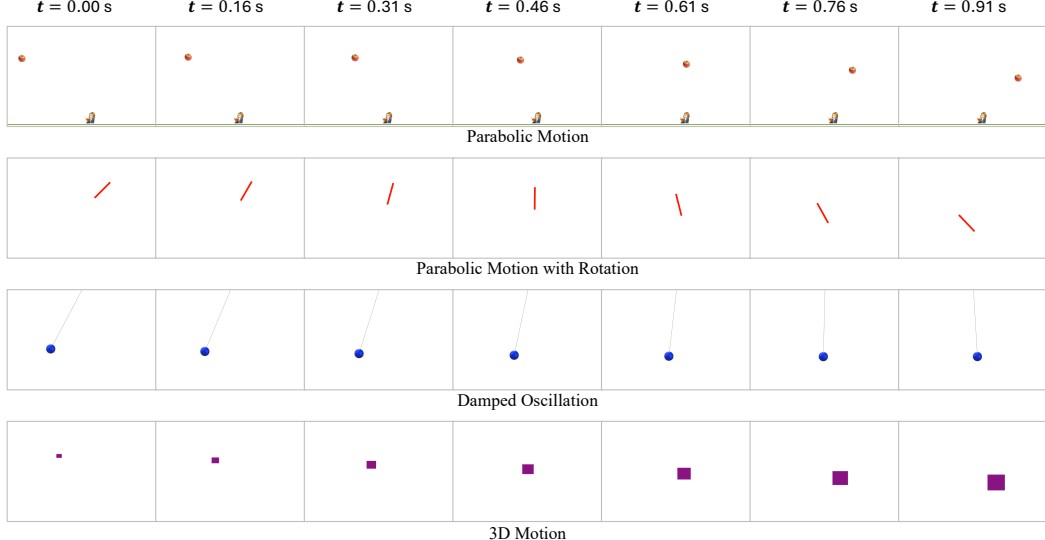

Figure 6: Sample physics-clean videos generated by our simulator.

## C  MORE DETAILS OF NEURAL NEWTONIAN DYNAMICS

### C.1  NEURAL NEWTONIAN DYNAMICS NETWORK

In Algorithm 1 we present the detailed architecture of the Neural Newtonian Dynamics network. We model the most salient dynamics using a physics-driven linear Neural ODEs, and augment it with a learnable MLP to capture nonlinear and unknown dynamics.

---

**Algorithm 1** Neural Newtonian Dynamics Network Architecture

---

**Require:** Initial physical state $\mathbf{Z}_0 = [x, y, v_x, v_y, \theta, \omega, s, l, a]$, time stamps $t_0, \ldots, t_T$
**Ensure:** Future Latent physical states $\mathbf{Z}(t)$
 1: Define learnable parameters:
- $(a_x, b_x, c_x), (a_y, b_y, c_y)$ for linear 2nd-order dynamics of $(x, y)$
- $(g/L, \gamma)$ for linearized pendulum or circular motion $(\theta, \omega)$
- $(\alpha_s, \beta_s), (\alpha_l, \beta_l), (\alpha_a, \beta_a)$ for 1st-order dynamics of $(s, l, a)$
- Residual scale $\epsilon$

 2: Define residual MLP: $\mathrm{ResMLP} : \mathbb{R}^9 \to \mathbb{R}^6$ (initialized to 0)
 3: **for** each time $t$ **do**
 4:     Split $\mathbf{Z} = [x, y, v_x, v_y, \theta, \omega, s, l, a]$
 5:     Compute linear dynamics:

$$a_x^{\mathrm{lin}} = a_x x + b_x v_x + c_x$$
$$a_y^{\mathrm{lin}} = a_y y + b_y v_y + c_y$$
$$d\theta/dt = \omega$$
$$d\omega^{\mathrm{lin}}/dt = -(g/L)\theta - \gamma\omega$$
$$ds^{\mathrm{lin}}/dt = \alpha_s s + \beta_s, \quad dl^{\mathrm{lin}}/dt = \alpha_l l + \beta_l, \quad da^{\mathrm{lin}}/dt = \alpha_a a + \beta_a$$

 6:     Compute residual correction:

$$[a_x^{\mathrm{res}}, a_y^{\mathrm{res}}, d\omega^{\mathrm{res}}, ds^{\mathrm{res}}, dl^{\mathrm{res}}, da^{\mathrm{res}}] = \epsilon \cdot \tanh(\mathrm{ResMLP}(\mathbf{Z}))$$

 7:     Update derivatives:

$$\frac{d\mathbf{Z}}{dt} = [v_x, v_y, a_x^{\mathrm{lin}} + a_x^{\mathrm{res}}, a_y^{\mathrm{lin}} + a_y^{\mathrm{res}}, d\theta/dt, d\omega^{\mathrm{lin}} + d\omega^{\mathrm{res}}, ds^{\mathrm{lin}} + ds^{\mathrm{res}}, dl^{\mathrm{lin}} + dl^{\mathrm{res}}, da^{\mathrm{lin}} + da^{\mathrm{res}}]$$

 8: **end for**
 9: Integrate ODE by odeint to obtain $\mathbf{Z}(t)$ over $t_0, \ldots, t_T$

---

### C.2 ACCURACY OF NEURAL NEWTONIAN DYNAMICS PREDICTIONS

Figure. 7 to Figure. 18 show the predictions of the trained Neural Newtonian Dynamics (NND) model for each type of motion. Given the initial physical state $\mathbf{Z}_0$, the model's predicted physical states closely follow the ground truth over time.

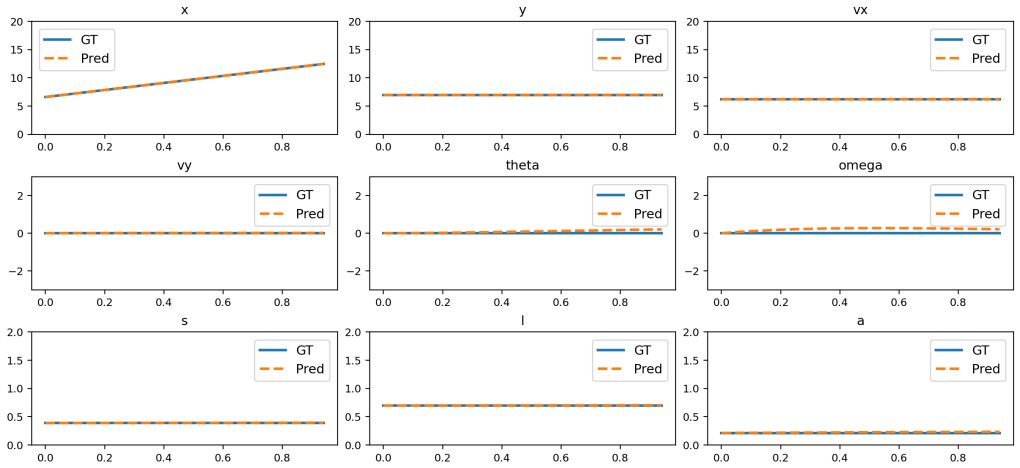

Figure 7: Comparison of NND predictions and ground truth for uniform motion.

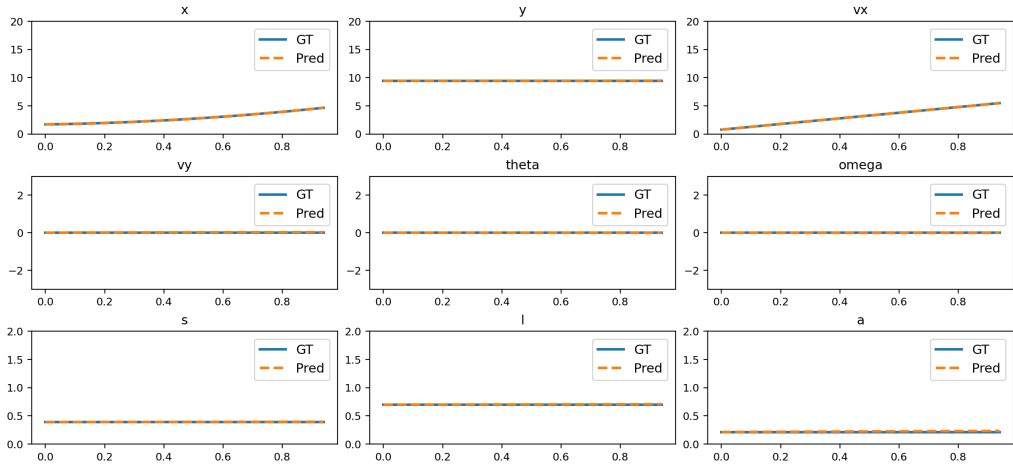

Figure 8: Comparison of NND predictions and ground truth for acceleration.

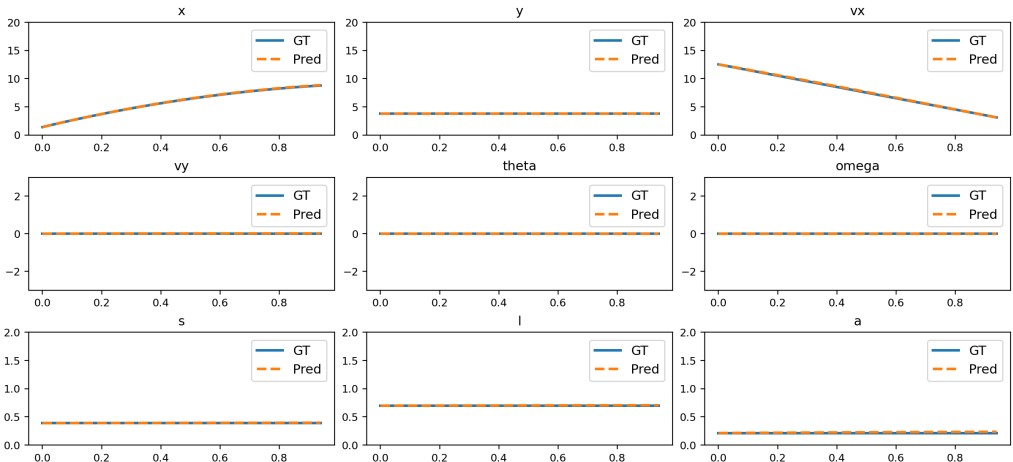

Figure 9: Comparison of NND predictions and ground truth for deceleration.

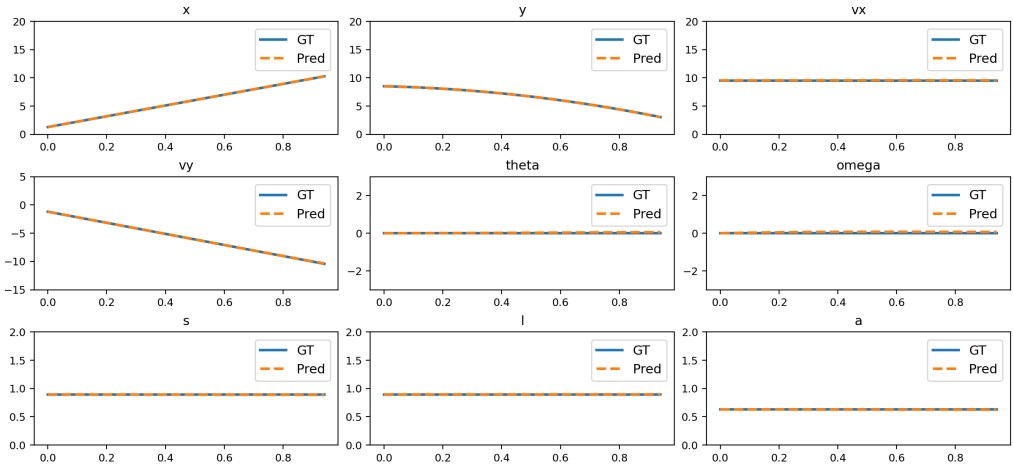

Figure 10: Comparison of NND predictions and ground truth for parabolic motion.

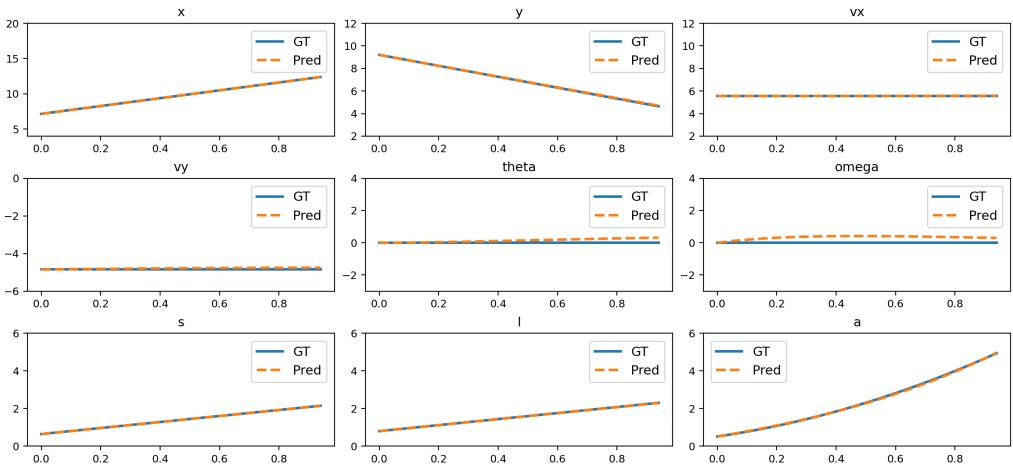

Figure 11: Comparison of NND predictions and ground truth for 3D motion.

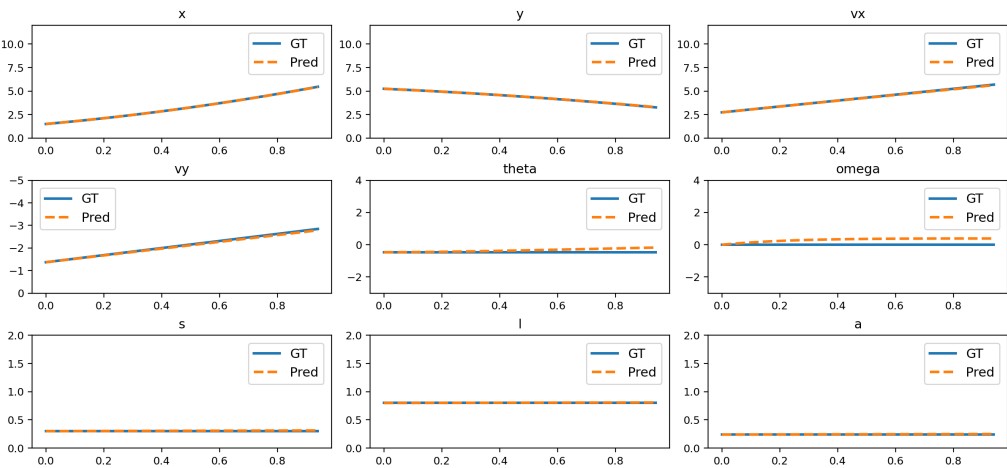

Figure 12: Comparison of NND predictions and ground truth for slope sliding.

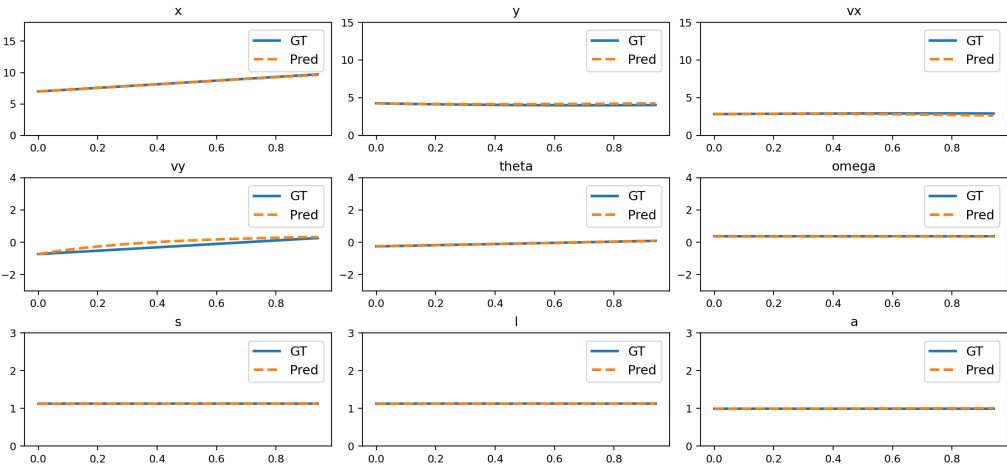

Figure 13: Comparison of NND predictions and ground truth for circular motion.

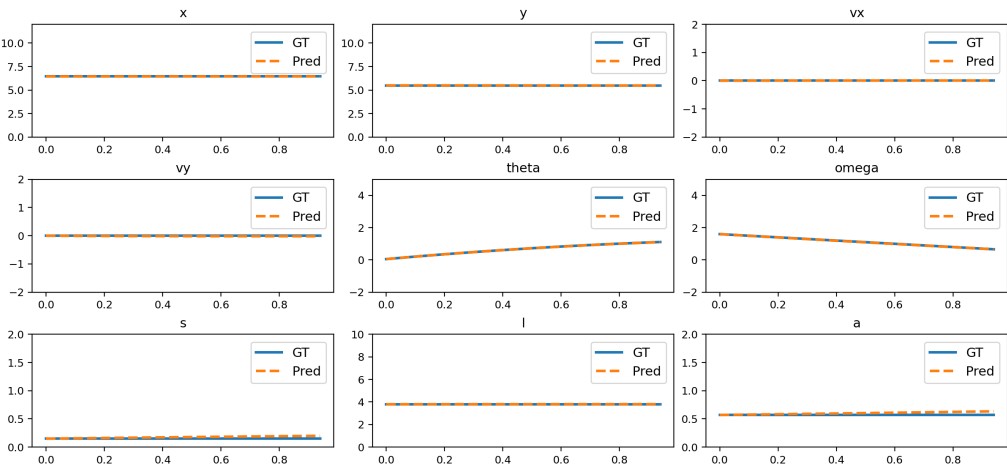

Figure 14: Comparison of NND predictions and ground truth for rotation.

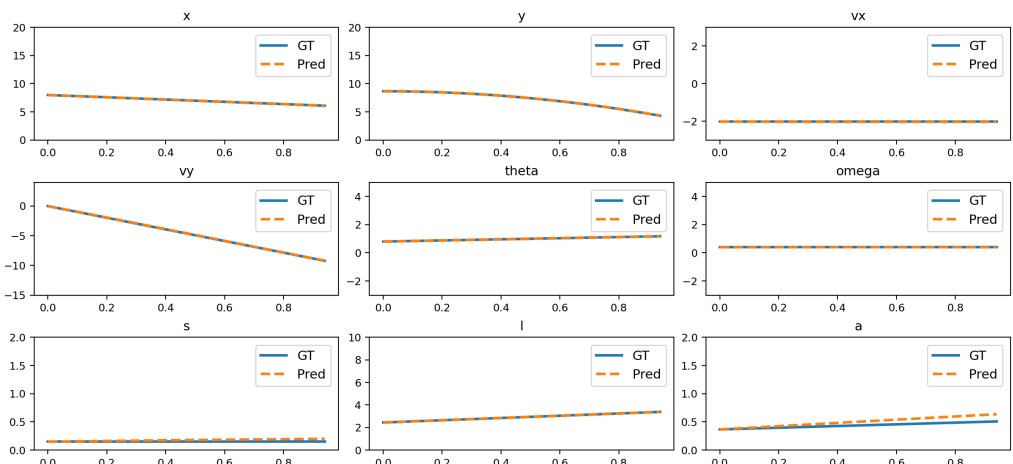

Figure 15: Comparison of NND predictions and ground truth for parabolic motion with rotation.

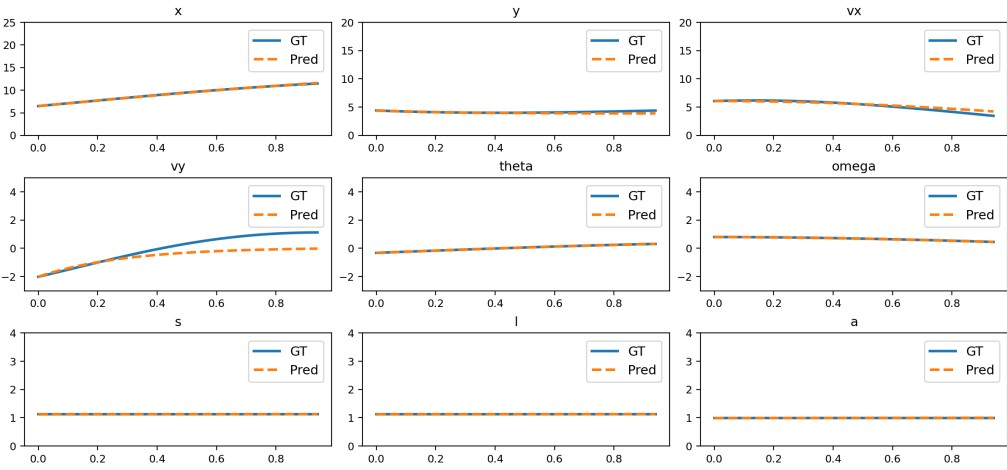

Figure 16: Comparison of NND predictions and ground truth for damped oscillation.

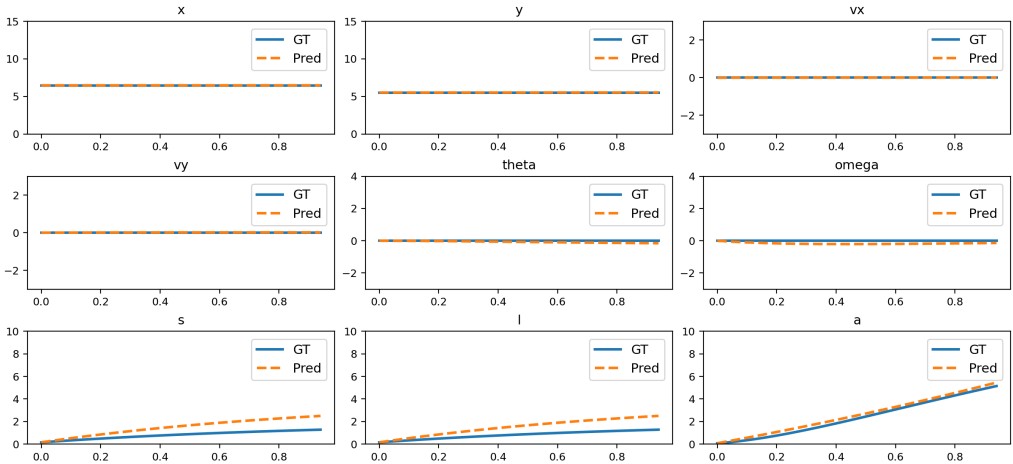

Figure 17: Comparison of NND predictions and ground truth for size changing.

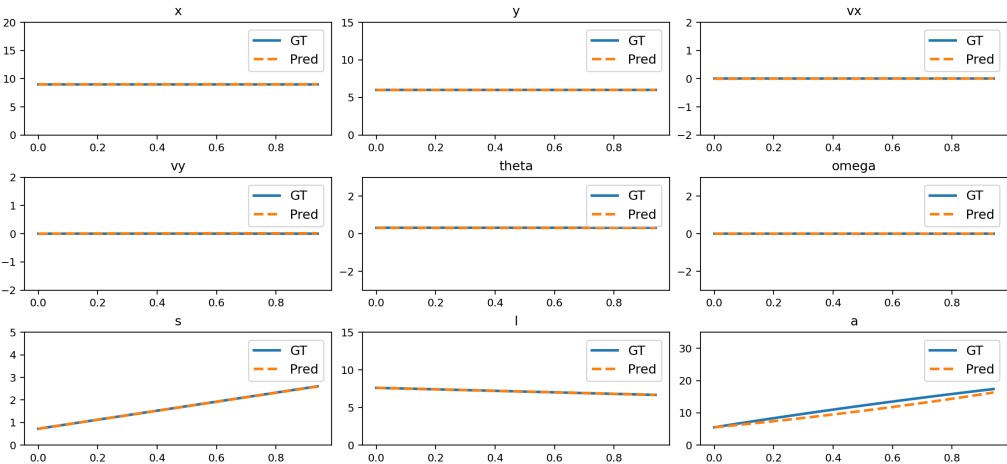

Figure 18: Comparison of NND predictions and ground truth for deformation.

# D  EVALUATION DETAILS

## D.1  PHYSICAL INVARIANCE SCORE

The Physical Invariance Score (PIS) as described in equation 7 indicates whether a certain quantity $C$ remains invariant over time. If the laws of physics are replicated perfectly, $C$ remains constant, and $C_\sigma \to 0 \implies \text{PIS} \to 1$. For each type of motion, a suitable $C$ should be selected.

**Uniform Motion**: An object is prompted to travel horizontally in uniform velocity in each scene. Therefore, we select the horizontal velocity $v_x$ as the invariant feature.

**Uniform Acceleration and Deceleration**: Under these motions, we check if the object obeys the law of accelerating (or decelerating) at a constant rate. The guidance parameters and prompts specify horizontal motion. Therefore, we set $C = a_x$ for acceleration, and $C = -a_x$ for deceleration.

**Parabolic Motion**: Under this motion, there is no horizontal acceleration. Therefore $v_x$ is expected to be constant. Additionally, the vertical acceleration $a_y$ due to gravity should be constant.

**3D Motion**: The prompt guides an object to travel towards the observer, creating the effect of increasing object dimensions, while also having a 2D motion. We approximate this effect as having a constant vertical velocity $v_y$, and a constant increment rate in the long-axis of the object $\Delta l$.

**Slope Sliding**: An object is prompted to slide down a constant slope. Assuming negligible effects from friction, we can expect accelerations $a_x, a_y$ to be constant.

**Circular Motion**: Objects are guided to orbit in a circular path, we assume the angular velocity about the orbital center $\omega$ is constant.

**Rotation**: When objects are prompted to "spin" or "rotate about their axes", we assume that the brief duration of the video, that they rotate in a constant angular velocity $\omega$.

**Parabolic Motion with Rotation**: Videos under this category should describe a superposition of a projectile motion under gravity, and a rotation about the object's axis. Therefore the metrics used in these two motions $(v_x, a_y, \omega)$ are used for $C$.

**Damped Oscillation** is simulated through various instances of pendulums, hinged at the top. We assume small angles $(\theta)$ for the stride. This leads to the vertical force varying with $cos(\theta)$, and we assume it to be a constant. Thereby we use $C = a_y$.

**Size Changing**: We prompt videos where it's natural to increase an object's overall size, while maintaining it's aspect ratio.(e.g., an inflating balloon). Assuming a constant rate of inflation, we set $C = \Delta r$; the rate of increasing the radius of the object.

**Deformation**: Objects under this category should expand, stretch, or spread-out over time. The aspect ratios may change. (e.g., the spread of a thick viscous liquid). We assume that the object increases its dimensions at a constant rate, and track this rate along it's long axis $\Delta l$.

After selecting $C$ for a motion type, $C_\sigma, C_\mu$ is calculated for every video. This lends to a PIS score per video. The final score reported in table 1 show the median PIS score after generating 12 different videos for each motion. In our case, temporal derivatives are formed from successive frames with $\Delta t = 1/(\text{FPS value})$, then mapped to physical units using a constant of 0.00625 meters per pixel. Each video is preprocessed with a 5-frame moving-average filter to reduce noise in derivative estimates.

Some feature assumptions are idealizations that may not hold in the real world. Accordingly, we report a **Reference** PIS computed directly from the guidance mask used to drive the video generation. For example, in 3D motion, $\Delta l$ need not be constant, so expecting $C_\sigma = 0 \implies \text{PIS} = 1$ is unrealistic. The mask-based PIS instead, serves as a practical upperbound– the score that would be achieved if the generator perfectly followed the guidance mask (which itself has $C_\sigma \neq 0$).

Table 3: Samples of Testing Prompts.

| Motion | Testing Prompts |
| --- | --- |
| Uniform Motion | A small metal cube sliding steadily along a smooth laboratory bench, reflections visible on the surface, scattered tools in the background, captured from a fixed side camera. |
| | A red rubber ball rolling at constant speed on a polished wooden floor, pulled by a thin string, with scattered papers and books in the background, observed from a fixed side camera. |
| Acceleration | A red sedan accelerating in a straight line on a clean highway, the road flat and clear, with only a pale sky and distant horizon in the background, captured from a fixed roadside camera. |
| | A black off-road SUV accelerating in a straight line on sandy terrain, with continuous sand dunes in the background, a few white clouds in the sky, sunlight slanting, kicking up fine sand particles, viewed from a stationary side-angle camera. |
| Deceleration | A yellow bus decelerates in a straight line in front of a traffic light on a city street, with pedestrians crossing nearby, and the wet road reflecting the sky, captured by a fixed side-view camera. |
| | A red coach brakes and decelerates in a straight line on a highway, with road signs and streetlights nearby and the city skyline visible in the distance, captured by a fixed side-view camera. |
| Parabolic Motion | A golf ball is hit at an angle with an initial speed. The camera captures its parabolic trajectory from the side. The scene takes place on a sunny golf course with manicured fairways, sand bunkers, and distant trees, adding depth and realism. |
| | A volleyball is served at an angle, captured from the side by a stationary camera. The scene is set on an outdoor beach volleyball court, with sand texture, net, and distant palm trees in view. |
| 3D Motion | A fighter jet accelerates slowly from the distance along the runway towards the camera, hangars and runway lights visible in the background, captured from a fixed oblique side camera. |
| | A cardboard box slides from the distance along a warehouse floor towards the camera, shelves and crates visible in the background, captured from a fixed oblique side camera. |
| Slope Sliding | A hardcover book accelerating down a carpeted inclined board in a classroom, chalkboard and desks in the background, captured from a fixed side camera parallel to the ramp. |
| | A small metal cube sliding down a laboratory ramp, shiny reflections on its surface, scattered tools and wires in the background, captured from a fixed side camera parallel to the ramp. |
| Circular Motion | A tiny moon orbits a gas giant along a smooth, circular path. The top-down view shows the consistent motion without motion trails. |
| | A comet with a glowing tail orbits a distant star along a stable circular path. A top-down perspective emphasizes the symmetrical orbit and the stationary central star. |
| Rotation | A metal rod spinning on a concrete floor, faint scratches and dust visible, captured from a fixed top-down camera. |
| | A wooden dowel rotating gently on a tiled kitchen floor, soft shadows from ceiling lights, viewed from a stationary overhead camera. |
| Parabola +Rotation | A pen is thrown at an angle, rotating as it falls. Captured from a side camera, the notebook and desk provide background details and depth. |
| | A thin cylindrical rod gently tossed, rotating along its long axis, fixed side camera, realistic reflections, ground shadows visible, subtle motion blur. |
| Damped Oscillation | A small decorative bell hanging from a fine chain. The fixed camera captures realistic material and shadows. |
| | A realistic pendulum with a spherical bob swinging from a fixed pivot. The fixed camera captures the entire motion. |
| Size Changing | A red helium balloon gradually inflating in a sunny park, children playing in the background, trees casting soft shadows, captured from a stationary side camera. |
| | A transparent water balloon expanding in a laboratory, scientific instruments and glassware around, bright fluorescent lights overhead, captured from a fixed top-down camera. |
| Deformation | A long strip of yogurt slowly spreads into a smooth layer, captured by a fixed overhead camera. |
| | A long strip of jelly gradually deforms and flattens on a plate, captured by a fixed overhead camera. |

# E   MORE VISUAL RESULTS

## E.1   MORE GENERAL COMPARISON RESULTS

From Figure. 19 to Figure. 30, we provide additional visual results and comparisons with other methods.

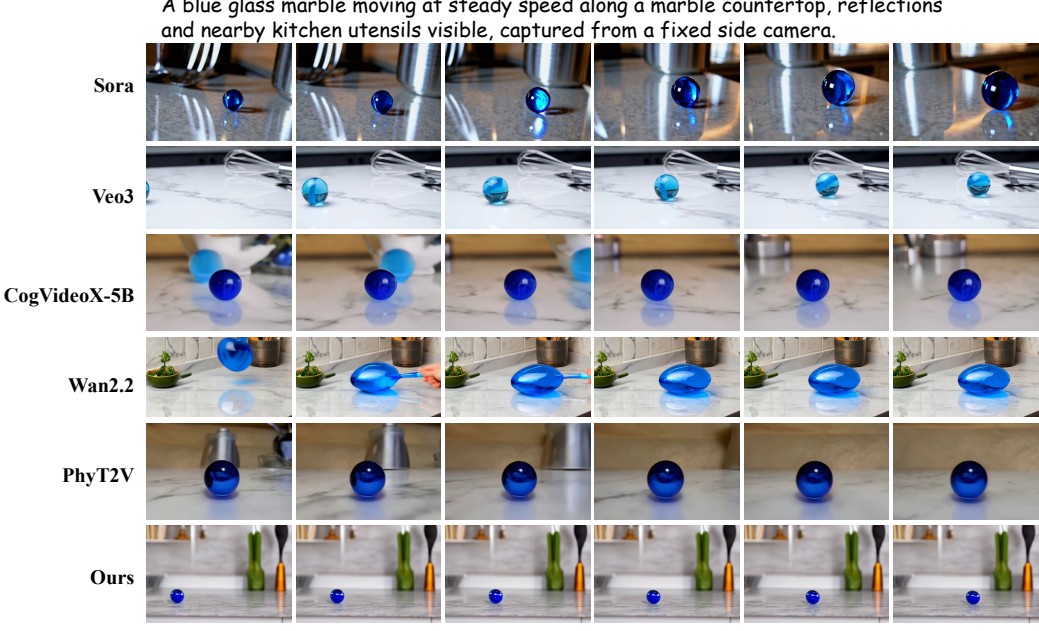

Figure 19: Visual comparisons on uniform motion.

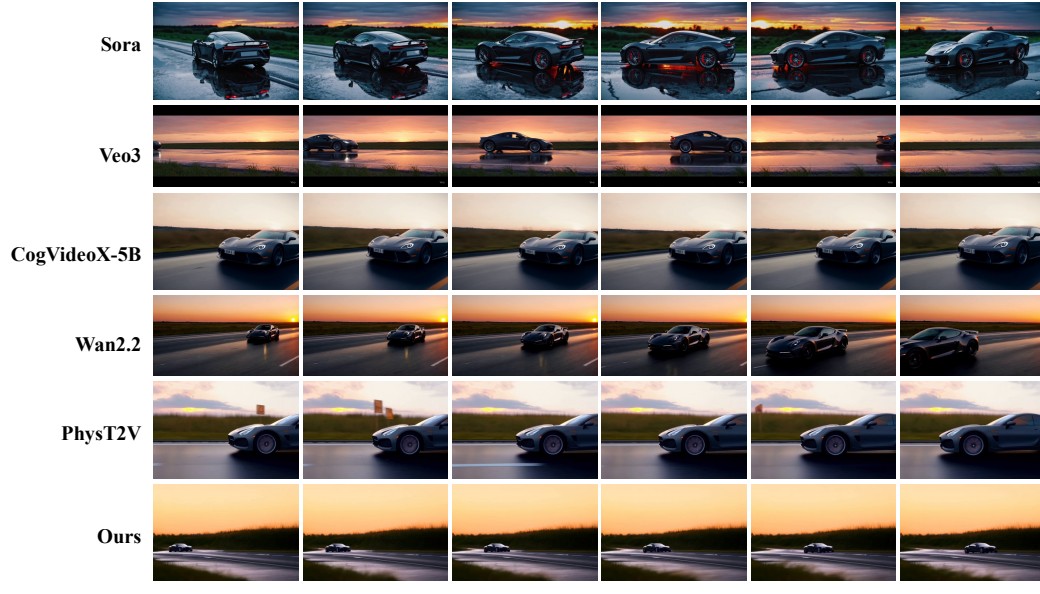

Figure 20: Visual comparisons on acceleration.

A red bowling ball rolls in a straight line on a wooden lane and gradually decelerates, slowing down before reaching the pins, with the polished lane surface reflecting light, captured by a fixed side-view camera.

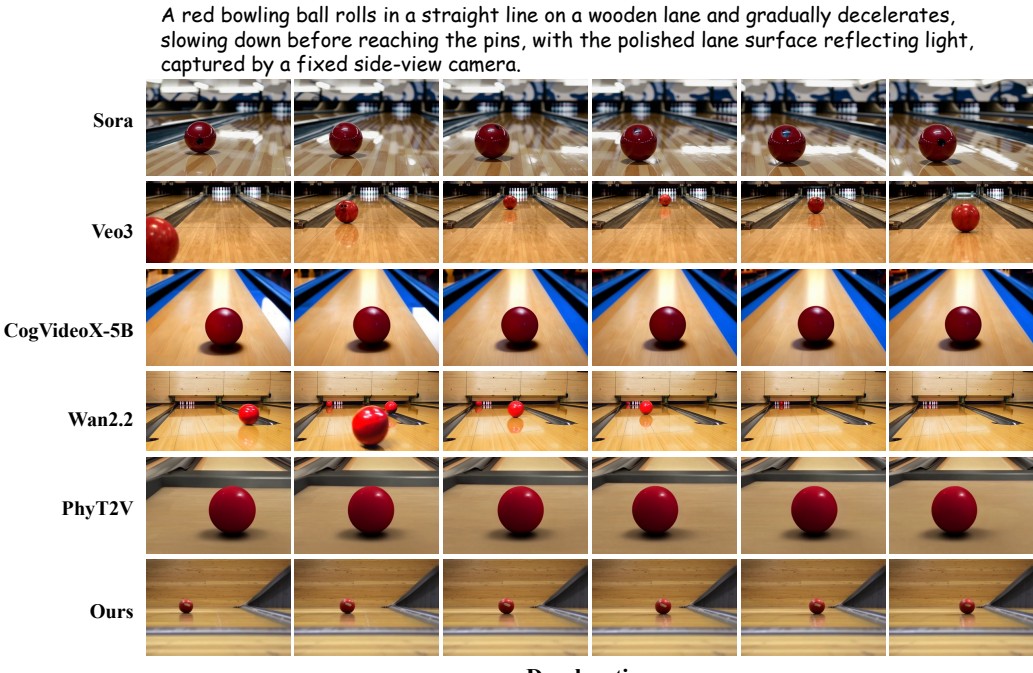

**Deceleration**

Figure 21: Visual comparisons on deceleration.

A baseball is thrown at an angle with an initial speed. The camera captures its flight from the side, rising and then descending. The scene is set on a baseball field, with dirt infield and green outfield grass, and stadium seats faintly visible in the background.

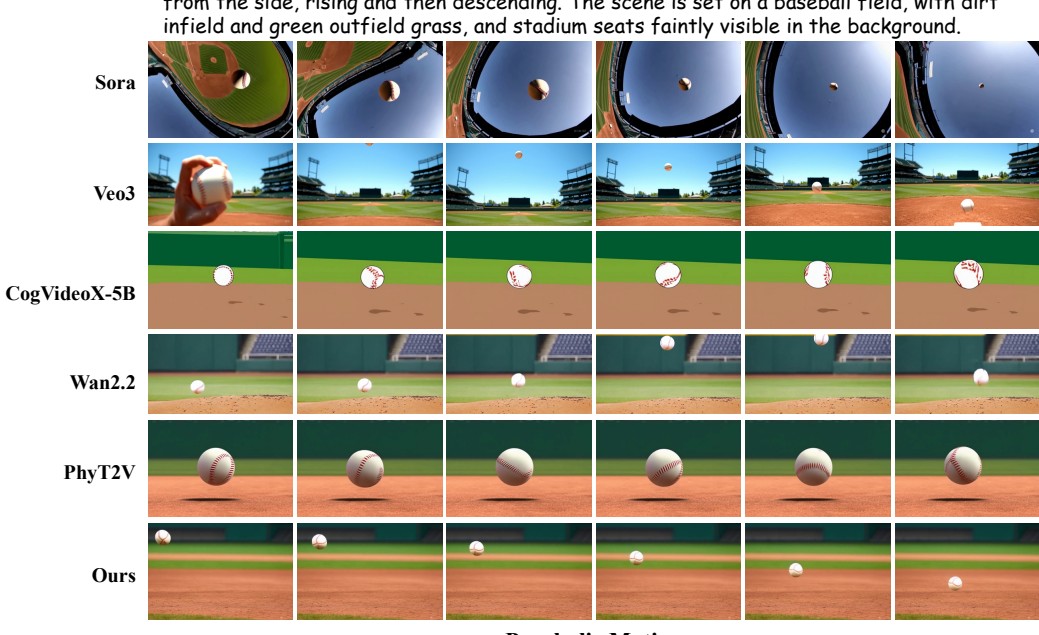

**Parabolic Motion**

Figure 22: Visual comparisons on parabolic motion.

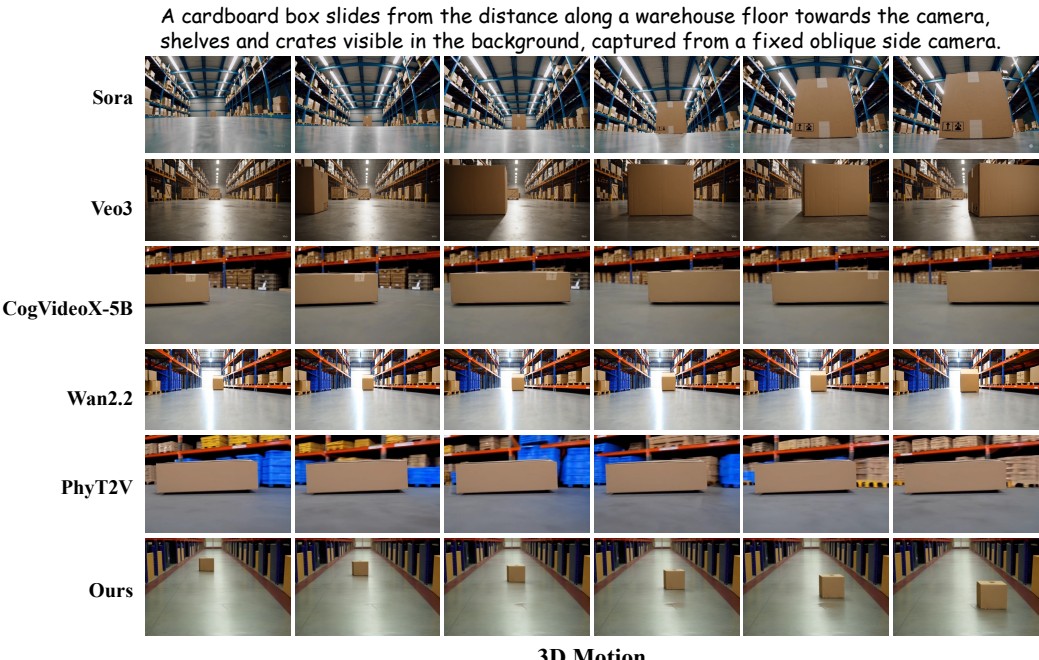

Figure 23: Visual comparisons on 3D motion.

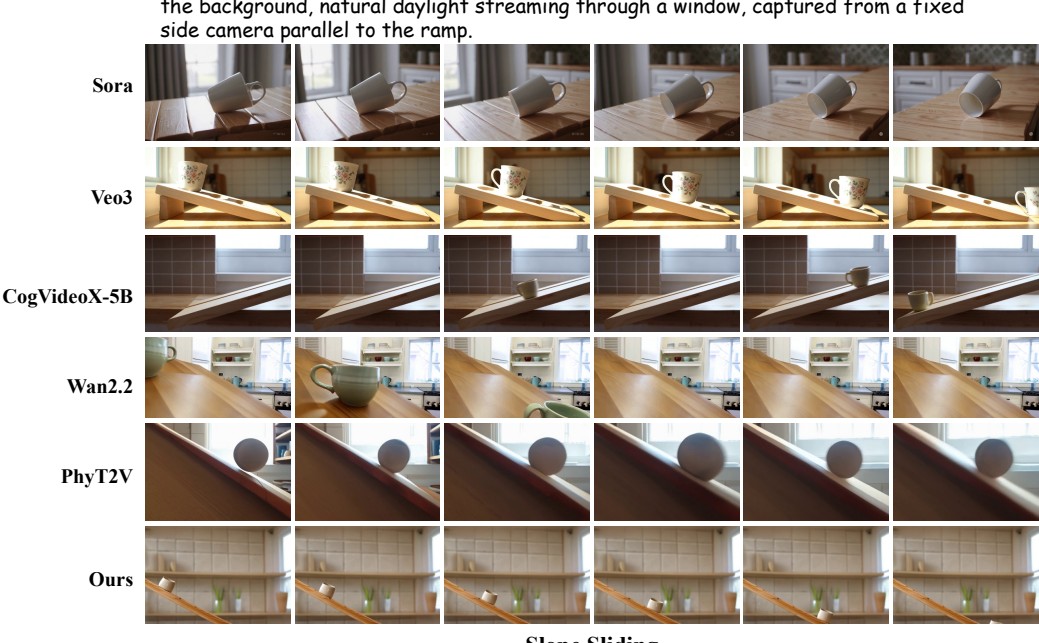

Figure 24: Visual comparisons on slope sliding.

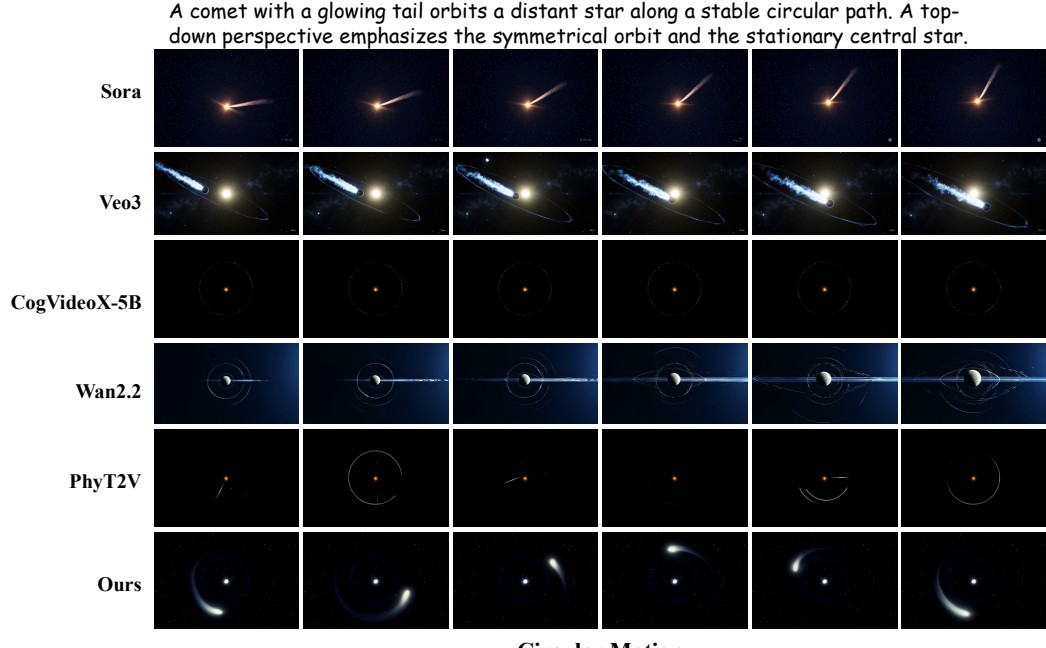

Figure 25: Visual comparisons on circular motion.

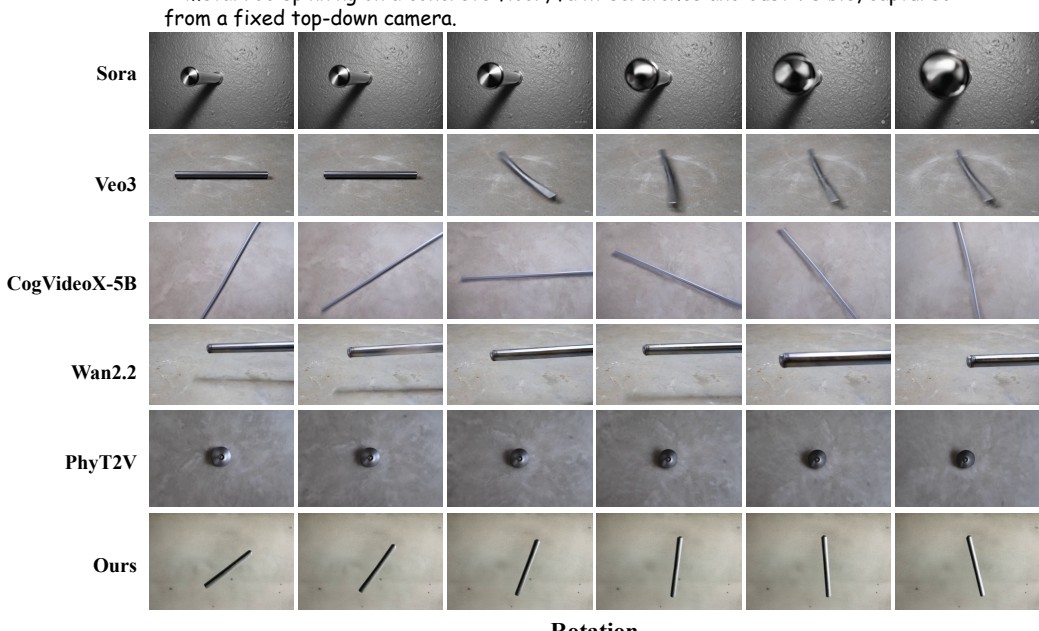

Figure 26: Visual comparisons on rotation.

A paintbrush is thrown at an angle, rotating while falling. Captured from a side camera, the artist's desk with palette and papers serves as background.

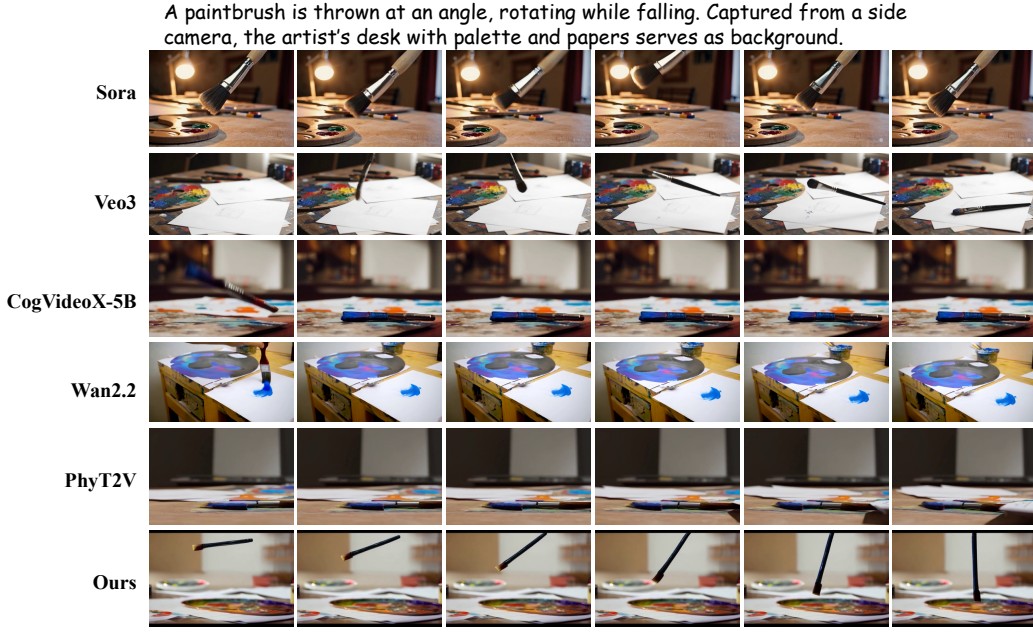

**Parabolic Motion with Rotation**

Figure 27: Visual comparisons on parabolic motion with rotation.

A small decorative bell hanging from a fine chain. The fixed camera captures realistic material and shadows.

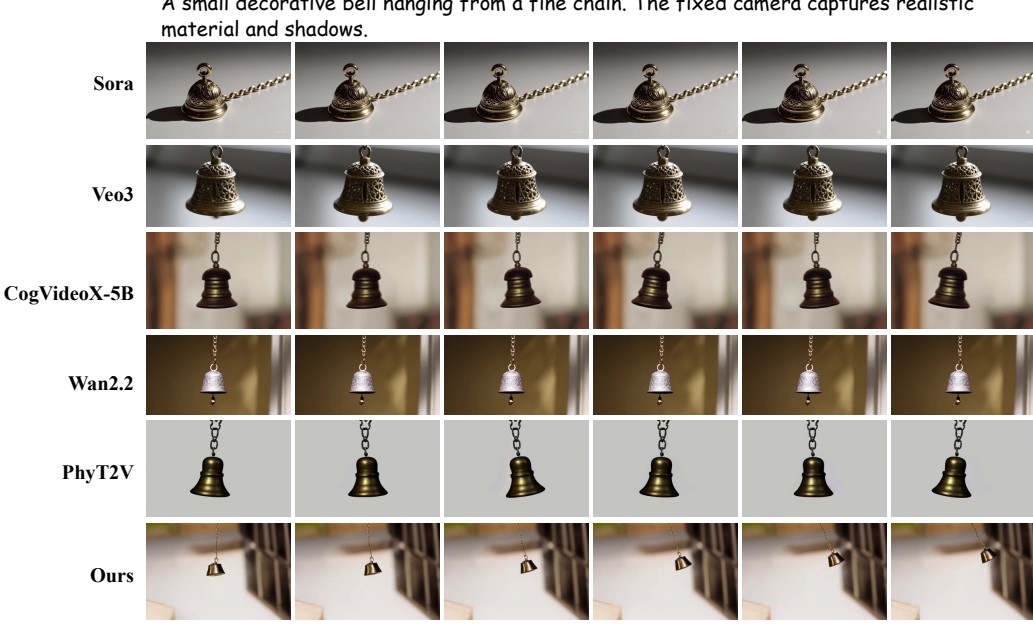

**Damped Oscillation**

Figure 28: Visual comparisons on damped oscillation.

A metallic silver balloon gradually expanding in a festive hall, fairy lights and streamers in the background, colorful reflections on the floor, captured from a stationary overhead camera.

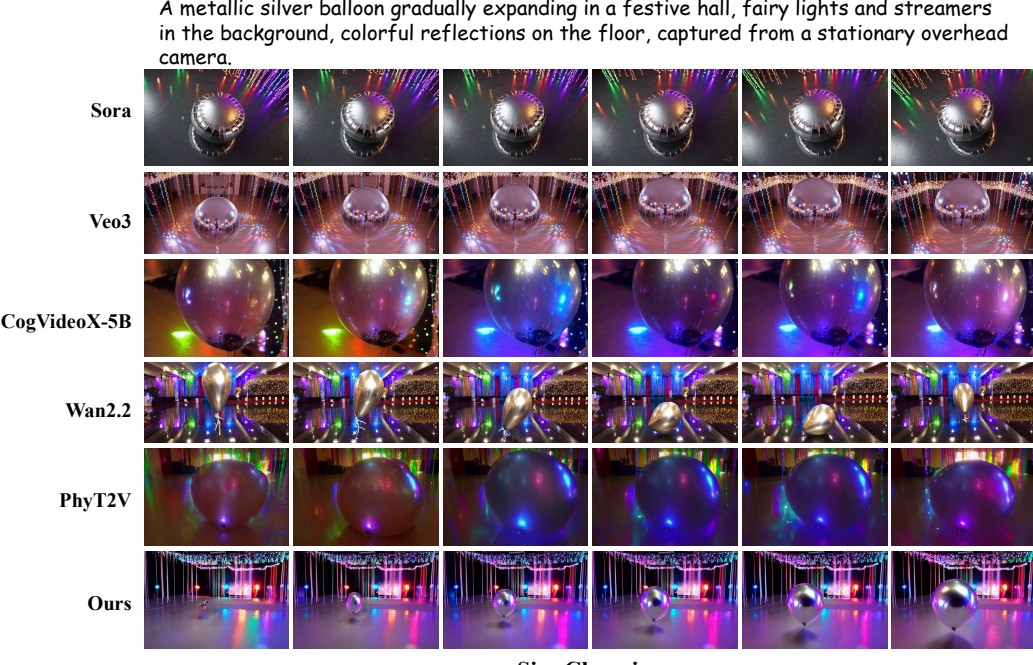

**Size Changing**

Figure 29: Visual comparisons on size changing.

A piece of soft dough is evenly flattened on a workbench, captured by a fixed overhead camera.

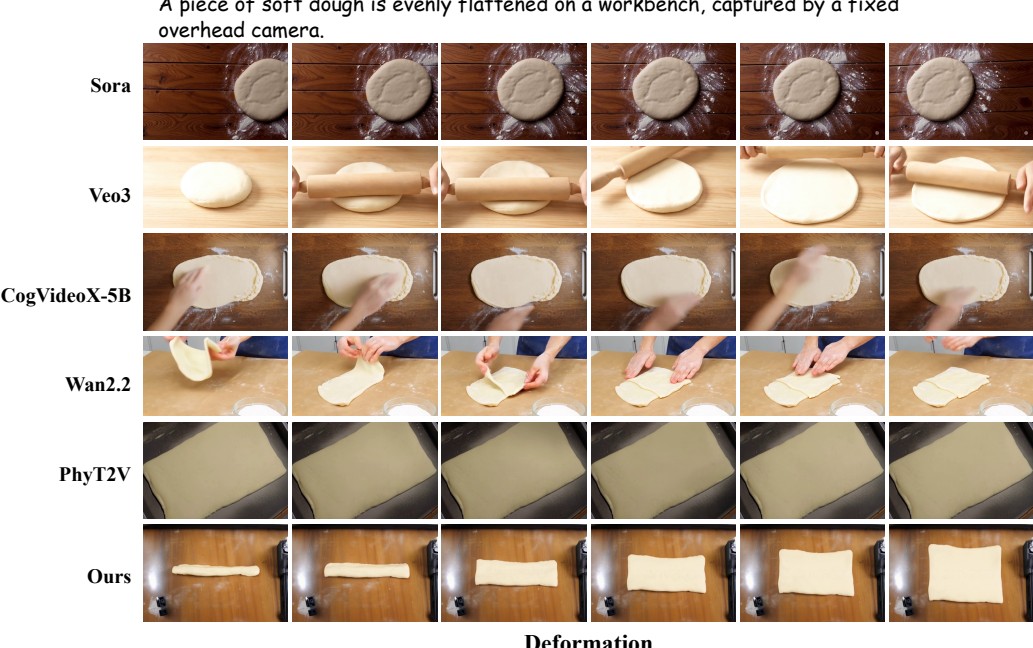

**Deformation**

Figure 30: Visual comparisons on deformation.

### E.2 MORE PARAMETER CONTROLLABILITY COMPARISON RESULTS

Figure. 31 and Figure. 32 illustrate the physical parameter control capability of NewtonGen.

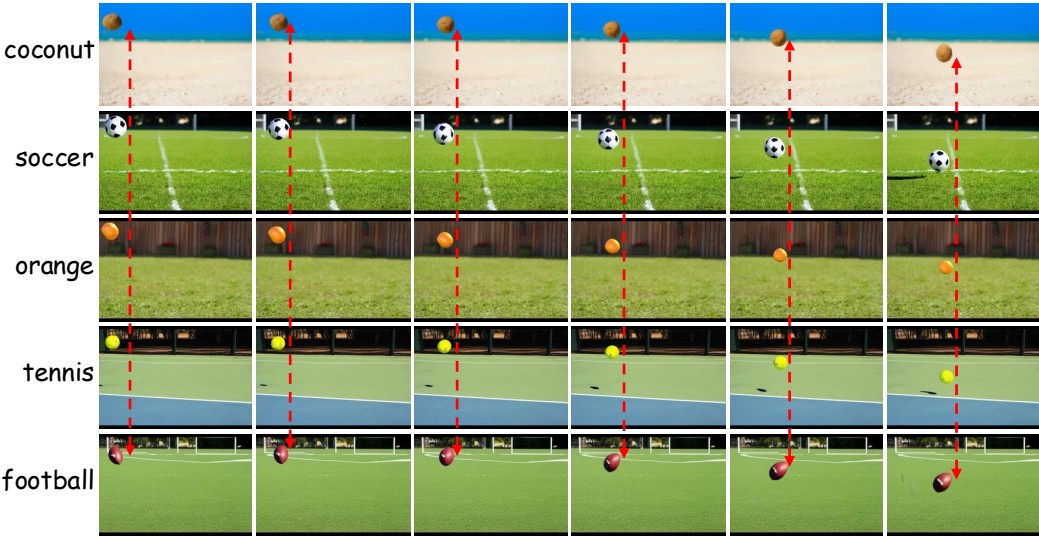

**Same Initial Location [x=2.0, y=10.0], Initial Speed [$v_x$=5.0, $v_y$=0.5]**

Figure 31: Given the same initial physical states but different scene descriptions, NewtonGen can generate diverse scenes with consistent motion.

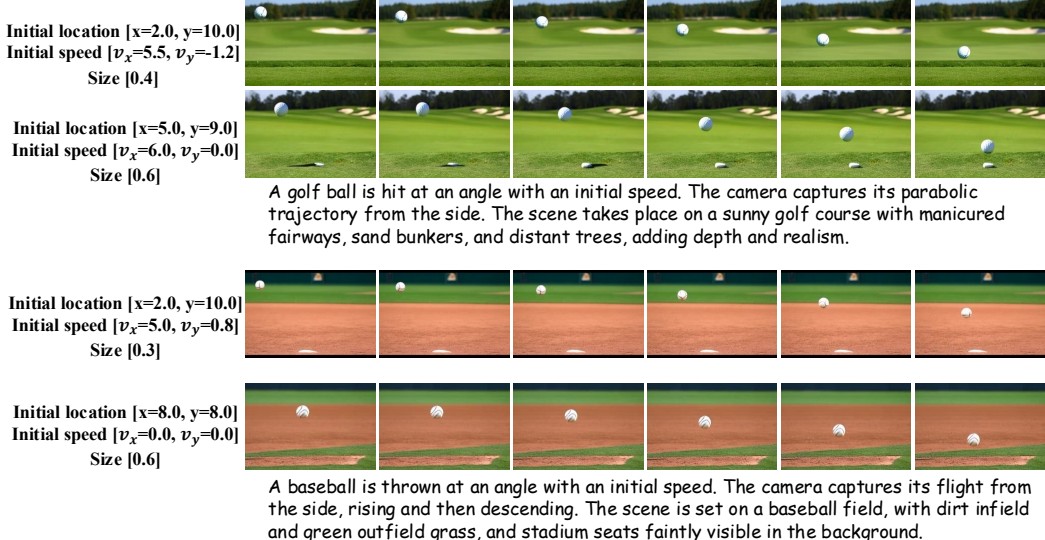

Figure 32: Given different initial physical states but the same scene description, NewtonGen can generate the corresponding motions.

# F  QUESTIONS AND ANSWERS

**Question 1:** Why is a NND (neural ODE) necessary to model/forecast Newtonian motion, and why not train a simple neural network to predict the coefficients of a parabola (for the parabolic trajectory)?
**Answer 1:** Our NND learns the underlying dynamics behind different systems, rather than merely fitting simple kinematics (trajectories) from data. They also provide a unified framework capable of representing diverse types of dynamics.

**Question 2:** For some motions, the underlying physical dynamics equations are already known, so why do we still need neural networks to learn dynamics?
**Answer 2:** Many complex or real-world motions are difficult to capture with simple physical formulas. For example, when rotation, parabolic motion, and even deformation occur simultaneously, it is challenging for humans to explicitly formulate the underlying physical laws. In contrast, our ODE model directly learns the dynamics from video data.

**Question 3:** Does your physical control model compromise the generative model's original physical effects or performance (e.g., shadows)?
**Answer 3:** Empirically, we have not observed any degradation in physical plausibility, such as shadow dynamics, after applying control. Our framework is training-free in the second stage, it injects physically consistent optical flow as a control condition only during inference, which preserves the model's original capabilities.

**Question 4:** Can NewtonGen (NND) handle video generation tasks involving collisions, rebounds, or explosions?
**Answer 4:** Currently, NewtonGen (NND) does not support such cases, as it is designed for continuous dynamics. These tasks would require additional event-based ODEs or hard-coded implementations.

**Question 5:** Can NewtonGen generate the motions of multiple objects' motion in a video?
**Answer 5:** Yes. NND can independently predict the physical states of multiple objects and then feed them into the motion-controlling video generator. The main bottleneck for video quality lies in the latter.

**Question 6:** Why choose "Go-with-the-Flow" instead of other motion control models as the base model for the second-stage video generation?
**Answer 6:** Other models often control motion through trajectories or bounding boxes, which makes it difficult for them to handle tasks involving deformation or rotation. In contrast, Go-with-the-Flow is based on optical flow control and thus has the potential to address such challenges.

**Question 7:** Is NND fast during training and inference?
**Answer 7:** Yes. NND is trained in the latent space rather than directly on videos, and its learnable parameters are concentrated in a lightweight three-layer MLP. As a result, inference can achieve real-time or faster speeds.

