# OpenReview forum: "NewtonGen: Physics-consistent and Controllable Text-to-Video Generation via Neural Newtonian Dynamics"
_ICLR.cc/2026/Conference — ICLR 2026 Poster_

### Official Review · Reviewer_CpA9 · 2025-10-19

**Soundness:** 2
**Presentation:** 3
**Contribution:** 2
**Rating:** 2
**Confidence:** 4

**Summary:**

The paper presents a video generation pipeline, that consists on two models: 1) single-object motion generator; and 2) (frozen) foundational video diffusion model. The motion generator is parametrized to take the initial physical state of the object (position, rotation, size, velocity, etc) and predict the future ones. It is trained on a small synthetic dataset. The predicted states are then converted to an optical flow, and the video generation backbone is conditioned to generate the video based on it using the go-with-the-flow noise tweaking method. The pipeline is converted with existing open and closed source baselines and shown to generate physically more plausible . The evaluation is performed using the Physical Invariance Score.

**Strengths:**

- The paper targets a very important research problem, directing more community's focus towards it.
- The writing and exposition is very clear, and also provides an interesting overview of the past literature
- The paper releases the source code, which is a good step toward ensuring reproducibility and transparency.
- The method compares to top-notch closed-source video generation baselines which makes the results more trustworthy. I even checked the Veo3 generation quality on the prompt "a baseball ball flying from left to right", and confirm that it fails to understand the prompt even for such a simple scenario in ~50% of the cases.

**Weaknesses:**

- I believe the approach is overly simplistic and inherently unscalable. It would not help to generate complex realistic motions (e.g., a person juggling jellies, a cup breaking against a wall, a squirrel jumping on a thin tree branch, human dancing, etc.), which is the core physical challenge. In its current form, it is just a way to generate simple optical flow for a single object motion. Even single-object deformations (e.g., see example with the dough) are not generated realistically. The authors should clearly explain in detail how the approach could be scaled to very difficult motion cases described above for which it feels infeasible to 1) construct the pre-training dataset, and 2) create the initial physical state. I think that for simple rigid motion, it's not that modern video generators cannot render it, they just fail to understand the prompt.
- If one treats the paper not as a general-purpose pipeline which is improving physics in video generators, but as an "optical flow conditioning" paper, then there is not much novelty in it. There are dozens of work that condition video models on motion, and what the current pipeline is doing is automating optical flow generation (in an overly simplistic way) instead of asking a human to provide manual input. But this could be automated in a simpler ways without the need to train a model (e.g., by asking LLMs to generate the trajectories, either directly or writing a trajectory generation code).

**Questions:**

- It is not well described how the object attributes are computed using the "morphological analysis and OpenCV-based tools" (L295). Would be good to include more details for completeness in the paper (without requiring a reader to study the code).
- Type on L161: "equation, This"
- "most common physical motions in daily life (e.g., flying balls)": I disagree with this statement, and believe that the most common physical motion in daily life are humans moving and talking.

---

> ### Author Response · Authors · 2025-11-21
>
> We sincerely thank reviewer **CpA9** for the feedback. We will revise the paper accordingly and respond to the main points below.
>
> ---
>
> ### W1. **“simplistic and unscalable”**
>
> We believe there may be a misunderstanding here, and we appreciate the opportunity to clarify.
>
> Our model is **not** an oversimplification of the problem, nor is it unscalable. We kindly encourage the reviewer to first check out the supplementary videos and comparisons, which illustrate better the difficulty of physically grounded video generation and the capabilities our method can achieve.
>
> We appreciate the reviewer’s opinion, and we fully acknowledge the value. However, we believe that this is more likely a **perspective issue than a technical weakness**. Different researchers may focus on different aspects of “core physical challenges.” We are fully aware of the importance of human and animal motion, and we totally agree that they are importantly. But our work targets the more fundamental classical mechanics. This is a distinct but equally important problem domain.
>
> Regarding the comment *"Even single-object deformations (e.g., the dough example) are not generated realistically,"* , our intention in Fig. 3 (last row) is to demonstrate precise physical decoupling and controllability—the dough’s deformation follows the rectangular shape predicted by NND, not the near-circle shape from video priors. We will add clarifications to avoid ambiguity, please check L427 in the revised paper.
>
> We stress that NewtonGen is not designed to only output optical flow. It is a general neural dynamics-aware generation framework that can learn and generate various types of dynamics. For more complex behaviors such as articulated human motion, the framework can in principle scale: one could decompose the body into joint-level components, learn an NND for each part, and assemble them during inference. This requires engineering effort but is fully consistent with our design philosophy.
>
> Regarding *"modern video generators can already render simple rigid motion; they just fail to understand the prompt,"* this conclusion differs from extensive findings in the community, for example
>
> * ICML 2025 – *How Far Is Video Generation from World Model: A Physical Law Perspective*
> * ICML 2025 – *PISA Experiments: Exploring Physics Post-Training for Video Diffusion Models*
> * ICML 2025 – *VideoJAM: Joint Appearance–Motion Representations*
> * ICCV 2025 – *VLIPP: Physically Plausible Video Generation with Vision–Language Physical Prior*
>
> These works unanimously confirm the difficulty of generating even basic rigid-body dynamics. The difficulty of physical generation goes far beyond understanding prompts, and the core challenge lies in the lack of effective physical representations and training objectives.
>
> ---
>
> ### **W2. “Optical flow conditioning”**
>
> The goal of our work is *not* optical flow conditioning, and we do not claim “optical flow controlled generatoin” in the paper. This is emphasized in the Preliminary Concepts section for our high-level motivation and idea.
>
> Our core contribution is a unified neural Newtonian dynamics framework that learns diverse physical dynamics from a small amount of physics-clean video data, and then integrates these dynamics with existing video-control models for physically consistent and controllable generation. It is **not our interest to concern about how to use optical flows/motions** to generative the final videos, since there are dozens of high-quality motion-controlled video. We care more how to learn/model all kinds of physical dynamics from data.
>
> Regarding *“asking LLMs to generate trajectories.”* as clarified in Sec. 2.2, the main limitation of this idea is the implicit assumption that large models already possess physical reasoning. In reality, both LLMs and current video models rely on data-fitting, and consistently fail on physically challenging, out-of-distribution scenarios. Moreover, generating trajectories with LLMs requires writing or modifying code separately for each motion type, each scene, and each physical configuration, which lacks scalability and general applicability. In contrast, NewtonGen learns different dynamics directly from videos and generalizes across scenes.
>
> ---
>
>
> ### **Q1. “More details about morphological analysis”**
>
> Thank you for this suggestion. We now provide additional details in the paper, please check L300.
>
> ---
>
> ### **Q2. “Typo at L161”**
>
> Thank you for catching this. We have corrected the typo.
>
> ---
>
> ### **Q3. “Most common physical motions in daily life”**
>
> We used flying-ball examples simply to provide readers with an intuitive understanding that many biological/mechanical/classical motions can be expressed using second-order ODEs.
>
> We again acknowledge the importance of human motion, but the goal of this paper is to articulate the Newtonian motion problem.
>
> ---
>
> We will incorporate all clarifications and improvements in the revised version. Thank you again for your time and comments.

---

> > ### Comment · Reviewer_CpA9 · 2025-11-25
> >
> > I am thankful to the authors for replying to my review. However, the response has not changed my perspective that the proposed method is just a toy optical flow generator in disguise of "fundamental classical mechanics" motion generator. I elaborate on this next.
> >
> > > We kindly encourage the reviewer to first check out the supplementary videos and comparisons,
> >
> > I *did* check the supplementary carefully before writing my original review, and I believe it was clear from it that I did. Moreover, I even went further and tried generating samples with Veo3 to confirm the author's original observation that modern video generators are extremely poor in generating simple consistent motion from text prompt. Accusations of superficial reviewing are unhelpful and do not promote constructive dialogue.
> >
> > > We are fully aware of the importance of human and animal motion, and we totally agree that they are importantly. But our work targets the more fundamental classical mechanics.
> >
> > There is a misunderstanding. I am not promoting doing human/animal motion generation. I am claiming that the method is unable to generate any realistic complex motion, because there is no realistic complex motion in its training dataset. And it's infeasible to construct a training dataset for the motion generator which would contain any realistic complex motion (one can trivially add something like brownian motion, but that is not useful).
> >
> > > The dough’s deformation follows the rectangular shape predicted by NND, not the near-circle shape from video priors.
> >
> > But that's exactly the problem! Dough does not deform just by its own in the way the NND predicts it to, and it should be human hands rolling it out with a rolling pin, but NND entirely handicaps the base video model ability to do this because of the optical flow conditioning. In this way, it's a step back for video modeling, not a step forward.
> >
> > > We stress that NewtonGen is not designed to only output optical flow. It is a general neural dynamics-aware generation framework that can learn and generate various types of dynamics.
> >
> > Theoretically yes, but practically no: if its output cannot be translated into something the base video model can take as an input, then there is no use of it.
> >
> > > One could decompose the body into joint-level components, learn an NND for each part, and assemble them during inference. This requires engineering effort but is fully consistent with our design philosophy.
> >
> > That's similar to what human motion generators (e.g., https://arxiv.org/abs/1808.07371) do, so why NND is a better choice? How can NND assemble the predictions for different joints in a consistent manner?
> >
> > > Regarding "modern video generators can already render simple rigid motion; they just fail to understand the prompt," this conclusion differs from extensive findings in the community.
> >
> > There is a misunderstanding. I intentionally used the word "render" instead of "generate": if the video model is conditioned properly (e.g., with an optical flow like the current work do), then it can properly render the desired user inputs. But if only provided with a text prompt, it does not understand and follow the user's intent correctly.

---

> ### Author Response · Authors · 2025-11-26
>
> Thank you for response. However, we want to re-state our rebuttals to solve the misunderstanding and concerns.
>
> ### 1. *“Accusations of superficial reviewing are unhelpful and do not promote constructive dialogue.”*
> We have never accused your review of being superficial. Our intention is only to kindly ask you to re-examine the supplementary videos, especially the parts demonstrating our advantages in physical consistency and generalizability across diverse dynamics. We hope the reviews **focus on the technical parts like other reviewers, rather than perspective issues**.
>
> ### 2. *“I am claiming that the method is unable to generate any realistic complex motion.”*
> There is no evidence supporting this claim. Complex motion can be decomposed into multiple elementary motion primitives. NewtonGen explicitly models and controls these fundamental physical dynamics, which form the basis of complex motions. To the best of our knowledge, no existing video generation model (whether open-source or closed-source) can produce physically realistic and controllable complex motion as the reviewer demands. Thus, such a requirement is not aligned with the current state of the field and is not reasonable at this stage.
>
> ### 3. *“And it's infeasible to construct a training dataset for the motion generator which would contain any realistic complex motion.”*
> Please refer to our new experiments showing that NewtonGen can also be trained on real-world datasets. We can learn underlying dynamics from real videos, and generate more physical videos. Moreover, realistic complex motions can be obtained from Blender or other physics-based simulators, or from real-world videos, and these can be used to train NND. This is a feasible approach. Our paper focuses on fundamental Newtonian mechanics because it provides the most essential and interpretable physical behaviors, not because complex data is infeasible.
>
> ### 4. *“Dough does not deform just by its own ... it's a step back for video modeling, not a step forward.”*
> In NewtonGen, physical dynamics and scene appearance are explicitly decoupled. NND models only the underlying physics, while the motion-controlled generator handles appearance and rendering. In the dough example, we deliberately used a rectangular proxy shape to highlight this decoupling property. This shape could just as well be elliptical or any other geometry (just like the real-world shape). Therefore, it is not justified to conclude from this single illustrative example that *“it is a step back for video modeling.”*
>
> ### 5. *“If its output cannot be translated into something the base video model can take as an input, then there is no use of it.”*
> Our paper presents over 12 distinct dynamic, demonstrating that NewtonGen effectively learns a wide spectrum of physical behaviors. The claim that *“there is no use of it”*  lacks supporting evidence and overlooks the core contribution of this work: **general-purpose neural dynamics modeling and learning**, which enables controllable physical evolution across conditions, initial states, and dynamic types.
>
> ### 6. *“That’s similar to human motion generators … why NND is a better choice? How can NND assemble predictions for different joints?”*
> The cited work (https://arxiv.org/abs/1808.07371) and NewtonGen operate under fundamentally different paradigms:
> The cited method performs **motion transfer (video-to-video translation)**: it transfers observed surface motions from one actor to another, and the result is strictly 1-to-1. NewtonGen performs **dynamics transfer**: instead of learning pose/trajectories, we learn the underlying dynamical laws. Once learned, NewtonGen can generate diverse outcomes under different initial conditions and configurations, while motion-transfer methods fundamentally cannot provide.
>
> Regarding joint consistency: NND learns local intrinsic dynamics from simulated or real data. These local dynamics naturally connect through the physical structure of the articulated body; ensuring structural connectivity is the role of the scene representation, not the dynamics learner.
>
> ### 7. *“If the video model is conditioned properly, it can render the user inputs. If only provided with text, it cannot understand the user’s intention.”*
> In fact, this is one of the motivations behind NewtonGen. We want to emphasize is that prompts are only a bridge between the semantic distribution and the video distribution. The real problem in physical video generation is not how well the model understands the prompt, but it’s that the video distribution itself lacks proper physical modeling.
>
> For example, in the ICML 2025 paper “How Far Is Video Generation from World Model: A Physical Law Perspective”, even when the first frame and physical setup are given, current models still fail to generate physical videos. This failure has nothing to do with prompt understanding.

---

> > ### Comment · Reviewer_CpA9 · 2025-11-26
> >
> > > There is no evidence supporting this claim [the method is unable to generate any realistic complex motion]
> >
> > There is no evidence disproving it either, and some basic logic supports it: there was not complex motion in the toy dataset the NND was trained on, it would be a miracle if it will suddenly be able to generate something complex (e.g., fire, water flowing, humans walking, etc.)
> >
> > > Please refer to our new experiments showing that NewtonGen can also be trained on real-world datasets.
> >
> > Do I get it right that this real-world dataset was then converted to a toy dataset with measurable physical charatectistics (using some off-the-shelf optical flow / segmentation models or manual masks) to train the NND? If that's the case, then again, it will be impossible to learn motion in general case for which it's impossible to extract these physical characteristics.
> >
> > > [Existing human motion generator comparisons]
> >
> > Oh, I provided the completely wrong reference and I apologize for that. What I meant is something like this (https://nvlabs.github.io/PhysDiff/), i.e. where there is some generative backbone synthesizing human motion, not a motion transfer model of course.
> >
> > > [Regarding the dough example] NND models only the underlying physics
> >
> > But hands rolling a rolling-pin is also physics, I do not understand the proposed dichotomy of motion into "physical" vs "non-physical". NND must also be able to model the physics of hands rolling the dough, it is not something that can be discarded simply because [such kind of motion is not the focus of the work]. The proposed method severely handicaps the generation capabilities of video models since they become unable to generate such stuff (i.e., when the optical flow from NND is provided).
> >
> > ------------------------------
> >
> > I feel like this discussion is not going anywhere, let's try structuring it around my main critics and your main argument against it. My main critics is that I am convinced that this approach is inherently unscalable . From what I understood, your main arguments against this are that 1) I cannot claim this because I have not tried; and 2) in theory, one can compute the physical characteristics of anything (given enough time), hence one can train an NND on it. Is that correct?

---

> ### Author Response · Authors · 2025-11-26
>
> Thank you for your timely reply, and we really appreciate your time. We also hope that you can evaluate our work from the research motivation and the problem we aim to address, so the contribution of NewtonGen to the community can be better understood.
>
> ### 1. *“There is no evidence disproving it… complex motion…”*
> We never claimed that our method is designed for complex motion in the paper. However, your claim “the method is unable to generate any realistic complex motion” is an conclusion, which we believe is not appropriate.
>
> We have now clarified NewtonGen’s scope and primary goal in L229. What we are modeling is fundamental Newtonian dynamics, not surface-level complex motions. If a model cannot learn basic dynamics, it cannot scale to more complex motions. We fully agree that complex physical motion generation is important, but water, fire, fluid motion, or human/animal motion generation are not within the scope of this paper. These are meaningful and challenging long-term research goals for the community. Before reaching that end goal, many works, including NewtonGen, are exploring different feasible approaches. Progress must be step-by-step, and we hope this can be understood.
>
> ### 2. *“Real-world dataset training”*
> Yes, but the dataset is not a toy dataset (please check the real-world videos we provide in the Supplymentary Materials). It is captured from real-world videos. From these sequences we can reliably extract essential physical properties such as: position, shape, velocity, orientation and etc. These are sufficient for modeling many common types of motion. As future vision models (like SAM3, SAM3D...) improve, we expect to extract even richer physical quantities (e.g., metric depth, surface, material, mass), which will further expand NewtonGen’s applicability.
>
> ###  3. *“Existing human motion generator comparisons”*
> NewtonGen and PhysDiff target fundamentally different problems:
> (i). PhysDiff relies on a human-specific physics simulator, manually defined and not generalizable. NewtonGen learns neural dynamics directly from data, making it adaptable across different dynamical systems.
> (ii) PhysDiff generates human motion silhouettes (no texture or background). NewtonGen generates full video appearance, closer to real-world scenes (Fig. 3 and Fig. 5).
> Thus, direct comparison is not meaningful because the goals, approaches, and assumptions differ.
>
> ### 4. *“Regarding the dough example”*
> We are not separating the scene into “physics” vs. “non-physics”. Here, “physics” (more precisely dynamics) refers to the motion behavior of the foreground object (e.g., how the dough shape changes). Appearance (texture, lighting, background) is handled by the motion-controlled generator.
>
> Regarding “the physics of hands rolling the dough”: Please note that the prompt does not describe any hand interaction (please check the last row of Fig. 3). It describes a very soft dough spreading on the table on its own. This example is intentionally simple to show the decoupling between dynamics and appearance.
>
> ---
>
>
> ### 5. *“Main concern is NewtonGen’s scalability. ”*
> Our response is three-fold:
> (i) NewtonGen already handles multiple different dynamics in a unified framework, thus this itself is a form of scalability (we show 12 kinds of 2nd order dynamics from simulation data and 1 dynamics from real-world data).
> (ii) With learned neural dynamics, we can generate infinite videos with different initial conditions, objects, backgrounds, colors, textures and etc (please check the Fig. 1(b) and Fig. 4).
> (iii) With more diverse training data, NewtonGen can scale to more dynamics. In this paper we focus on basic dynamics obeying classical mechanics. As more physics-clean datasets become available, NewtonGen’s core idea naturally extends to more complex dynamics. We also see early evidence of this trend in our hybrid-motion experiments. For example, the hybrid motion in Fig. 1(a) and the **Parabolic Motion with Rotation** cases in Fig. 3 and Table 1 demonstrate that NewtonGen remains effective even when the underlying dynamics become more complex.
>
>
> ---
>
> ### 6. Cross Check
> To ensure we are aligned and have more constructive dialogue, could you clarify which of the following concerns you feel have been addressed so far?
>
> (1) Over simplistic (2) Optical flow conditioning (3) Morphological analysis details (4) Common physical motions in daily life
> (5) Dough deformation behavior (6) Comparisons with existing human motion generators (7) Prompt understanding
>
> Thank you again!

---

> ### Comment · Reviewer_CpA9 · 2025-11-26
>
> Thank you for your response. Regarding the cross-check:
> - *over-simplistic / common physical motions in daily life / dough deformation / human motion generation*: all these concerns are the same category of my non-scalability critics, where I was just trying to make my point clear in different ways. They are not resolved
> - *OF conditioning / prompt understanding*: those questions were related to whether I should treat the current submission as a novel optical flow conditioning method or some new video generator. You suggested the latter, so we can treat those concerns as resolved (for "prompt understanding", I feel we didn't understand each other, but at this point, it's better to just skip this discussion)
> - morphological analysis details: thank you for including these details, this concern is resolved.
>
> Regarding the scalability arguments:
> > "NewtonGen already handles multiple different dynamics in a unified framework, thus this itself is a form of scalability".
>
> The fact that it can handle 12 kinds of dynamics does not imply that it can scale to arbitrary difficult physical processes (e.g., fire, water, smoke, etc)
>
> > With learned neural dynamics, we can generate infinite videos with different initial conditions, objects, backgrounds, colors, textures and etc (please check the Fig. 1(b) and Fig. 4)
>
> I don't understand this argument: the resulted videos would be overly simplistic and limited in motion diversity. Having them will not help training a model to generate arbitrary motion
>
> > With more diverse training data, NewtonGen can scale to more dynamics.
>
> If the backbone NND architecture is expressive enough (which is an investigation on its own), then yes. However, it will be impossible to generate a diverse enough dataset with physics parameters available unless AGI will suddenly appear and do this for us. Computer graphics, in its 50+ years of existence, has still not arrived to faithful motion modeling even for humans (for which billions of dollars have been invested), to say nothing about other motion categories. If we admit that we cannot synthesize a good enough motion dataset, then we should try to estimate physical parameters from real videos. But this is also infeasible because, in general case, you would end up trying to estimate stuff like stiffness of each individual hair strand of a squirrel's tail jumping on a branch, materials of the tree bark, all the materials friction and so on (let me know if I need to elaborate on why it's practically impossible).
>
> --------------------------------------------------
>
> Regarding this statement: "We never claimed that our method is designed for complex motion in the paper."
>
> The "physical" challenge of video modeling is in being able to generate physically correct motion without breaking the base video modeling capability (e.g. "hands unrolling dough", where hands generation come from "base modeling capability", and dough unrolling generation comes from the proposed NND method), not in being able to generate simple physically correct motion in isolation. You cannot just ignore the fact that you sacrifice the base video modeling capability for the sake of being able to generate simple correct physical motion.

---

> > ### Author Response · Authors · 2025-11-27
> >
> > Thank you for your response.
> >
> > ---
> >
> > ### 1. On the notion of *“scalability”*
> >  We reject your premise about the notion of "scalability". Scalability is defined with respect to the scope of a method. When a method shows its full functionality within the scope it lives on, it has achieved its goal.
> >
> > Therefore, to answer your concerns:
> >
> > (1) the scope of this work is to address **Newtonian motion, not arbitrary dynamics.** You are asking us to try something out of our scope. Verbally harassing us on this does not invalidate the fact that our method works within the scope we define. How about we ask you this question: Do you agree that our method works better than any other competing methods within the scope of Newtonian dynamics. If you disagree, please give us one concrete example by telling us the prompt, the dynamics, and the video.
> >
> > (2)  An NND has intrinsic limits on what it can model. We already talked this in L229–L231 and L193–L197. We never claim that an NND can handle arbitrary dynamics without limit.
> >
> > ### 2.  *“the resulted videos would be overly simplistic and limited in motion diversity” and that “having them will not help training a model to generate arbitrary motion.”*
> > We do not agree with this characterization:
> > Table 1, Figure 3, and the videos in the supplementary material clearly demonstrate that NewtonGen can generate diverse and highly detailed videos within the Newtonian regime.
> > We never claim that NewtonGen is designed to “generate arbitrary motion”; this is again an expectation that lies outside the declared scope of the work.
> > Thus, labeling our results as “overly simplistic” overlooks the empirical evidence we provide and shifts the evaluation away from what the method is actually designed to do.
> >
> > ### 3.  *“it will be impossible to generate a diverse enough dataset with physics parameters available unless AGI will suddenly appear and do this for us.”*
> > We also disagree with this statement:
> >
> > (1) Handling all possible motions in the real world is not the goal of this paper.  NewtonGen focuses on basic Newtonian dynamics and a manageable subset of physical parameters that are relevant to those dynamics. we are talking about "more dynamics" not "all motions"
> >
> > (2) We do not need to model every possible physical parameter to make progress on controllable, physics-aware video generation. The claim that this is “impossible” unless we have AGI is speculative and exaggerates the requirements for meaningful physics-based progress.
> >
> > **The reviewer’s argument here effectively inflates the problem definition far beyond the paper’s actual goals and then criticizes the method for not solving that inflated problem.**
> >
> > ### 4.  *“You cannot just ignore the fact that you sacrifice the base video modeling capability for the sake of being able to generate simple correct physical motion.”*
> > Table 1 in our paper explicitly shows that: Adding physical constraints via NewtonGen does **not degrade** the underlying video model’s generation quality.
> > In fact, NewtonGen exhibits improvements in background consistency and motion smoothness compared to the baselines. Therefore, the claim that we “sacrifice the base video modeling capability” is not supported by our quantitative results and is contradicted by the evidence we present.

---

> > > ### Author Response · Authors · 2025-11-27
> > > **Confirm the scope of NewtonGen**
> > >
> > > We acknowledge the reviewer’s ambitious goal of addressing the entire universe of video generation within a single paper. We are only addressing Newtonian motion. We think that this is the root of the disagreement.
> > >
> > > Would it be acceptable to the reviewer if:
> > >
> > > (1) We define the scope of this paper to Newtonian motion. We do not consider fire and water and human motion (which is never the goal).
> > >
> > >  (2) We claim that our method is able to generalize within the scope of Newtonian motion. We will not claim anything about arbitrary motion. We make the limitation clear in this paper (L229–L231, L193–L197,  and L537-L540).
> > >
> > > (3) We have provided evidence in Figure 1, Figure 3, Figure 19-30, Table 1, Table 3, and Supplementary videos. This volume of evidence (Newtonian motion video generation) is substantially more than any of the published papers/projects in the past years ([Sora], [Veo3], [CogVideoX-5B], [Wan2.2], [PhyT2V]).
> > >
> > > We believe that the reviewer is an experienced researcher, and therefore has been on the author’s side. While we cannot resolve every conflict, we are wholeheartedly interested in finding the middle ground that is acceptable to the reviewer.  Are the above changes sufficient to the reviewer? If not, we are open to hear your suggestions.

---

> ### Comment · Reviewer_CpA9 · 2025-11-27
>
> > Verbally harassing us
>
> I do not see how I verbally harassed you, could you please point out to the exact phrase and I will apologize for this?
>
> >  Do you agree that our method works better than any other competing methods within the scope of Newtonian dynamics.
>
> Yes, I 100% agree with that, but the baseline methods (Wan2.2, CogVideo, Sora, PhyT2V, etc.) do not sacrifice the ability of generating other types of motion while doing so!
>
> > "Adding physical constraints via NewtonGen does not degrade the underlying video model’s generation quality."
>
> The model becomes unable to generate any motion beyond what is dictated by the optical flow from NND, which is only simple newtonian motion. How is it not a degradation in quality?
>
> > Would it be acceptable to the reviewer if: (1) We define the scope of this paper to Newtonian motion.
>
> No, I do not find this acceptable. I suggest escalating this to the AC, as the current discussion has run its course and we are simply repeating the same arguments in different wording. The current paper constrains general-purpose video modeling to just simple Newtonian motion, develops the method which can only generate simple Newtonian motion, and compares to general-purpose video models. I do not find this justified unless being able to generate simple Newtonian motion is valuable on its own. Otherwise, it is just PISA benchmark hacking.

---

> > ### Author Response · Authors · 2025-11-28
> >
> > ### 1. *“Verbally harassing”*
> > (1) This stems from a deliberate misrepresentation of the scope of our work. Neither our method, nor any existing method to the best of our knowledge, is currently capable of generating or controlling "arbitrary complex motion." This is a limitation of the entire field at the present stage, not specific to our work.
> >
> > (2) All of our responses to your concerns have been grounded in facts, quantitative metrics, and visual evidence (videos), and we have addressed your points one by one in good faith. On the other hand, many of your statements are unsupported claims expressed in an **overly subjective** tone, for example:
> >
> > * *“Accusations of superficial reviewing are unhelpful and do not promote constructive dialogue”*
> > * *“there is no use of it”*
> > * *“it is a step back for video modeling”*
> > * *“I feel like this discussion is not going anywhere”*
> > * *“benchmark hacking”*
> >
> > These remarks do not engage with the technical content of the paper and instead attack the work and our responses at a rhetorical level, which is not aligned with the spirit of constructive scientific review.
> >
> > ### 2. *“baseline methods … do not sacrifice the ability of generating other types of motion”*
> > You state that baseline methods (Wan2.2, CogVideo, Sora, PhyT2V, etc.) “do not sacrifice the ability of generating other types of motion while doing so,” but do you have evidence for this claim?
> > By contrast, Table 1 in our paper offers concrete quantitative evidence:
> >  for example, PhyT2V does not improve, and in many Newtonian dynamics cases even reduces the performance of its base model (CogVideoX-5B).
> >
> >
> > ### 3. *“NND, which is only simple newtonian motion. How is it not a degradation in quality?”*
> > We would like to reiterate that the applicability of NewtonGen is explicitly defined within the scope of Newtonian dynamics. Our goal is not to generate or evaluate “all possible motions,” but to demonstrate controllable and physically consistent behavior within the Newtonian regime.
> > All comparisons and evaluations in the paper are conducted consistently under this scope. We do not claim to handle complex, non-Newtonian phenomena, and therefore it is not appropriate to judge NewtonGen by performance on motions that lie outside its intended domain. Within its scope, NewtonGen improves both physical consistency and controllability, while preserving or improving visual quality (See Table 1 and Fig 3, Fig. 4).
> >
> > ### 4. *“escalating this to the AC... ”*
> > We appreciate and welcome AC oversight.
> >
> > ### 5. *“The current paper constrains general-purpose video modeling to just simple motion”*
> > We do not constrain general-purpose video modeling; rather, we add physics-aware dynamics control on top of an existing base video model (such as Go-with-the-Flow).
> > Following your line of reasoning, a base model like Go-with-the-Flow (CVPR 2025 oral), which only accepts optical-flow conditioning for generation, also lose *“scalability”* and *“constrain general-purpose video modeling to just simple motion,”* and thus be *“of no use”*/*“a step back for video modeling”*, correct?
> >
> > ### 6. *“I do not find this justified unless being able to generate simple Newtonian motion is valuable on its own.”*
> > This statement implicitly dismisses not only our work, but also **the value and progression of physics-aware video generation research recognized by the entire community.**
> > Works such as PhysGen (ECCV 2024), PhyT2V (CVPR 2025), PhysGen3D (CVPR 2025), VLIPP (ICCV 2025), and Force Prompting (NeurIPS 2025), similarly focus on limited or simple motion regimes, as this is a necessary and interpretable foundation before scaling to more complex physical phenomena. Reviewer CpA9’s statement therefore imply that these widely recognized works are *“over-simplified”* and *“non-scalable”* and therefore *“of no use.”* We believe this is not the position acknowledged by the research community.
> > Does reviewer CpA9 agree that all of these works, including NewtonGen, have value in advancing community-recognized dynamic learning and physical controllability? If not, could you also confirm whether your stance implies that these works lack value because they focus on foundational, simple motion regimes?
> >
> > ### 7. *“benchmark hacking”*
> > This is another unsupported allegation. We have:
> > (1) Released videos in the supplementary material,
> > (2) Provided full code for computing the Physical Invariance Score (PIS), and
> > (3) Used the same evaluation protocol across all methods in a transparent and reproducible manner.
> > All benchmarks are designed and reported in a fair, consistent, and honest way. Labeling this as “benchmark hacking” without evidence is not justified.

---

### Official Review · Reviewer_UcEn · 2025-10-30

**Soundness:** 3
**Presentation:** 3
**Contribution:** 3
**Rating:** 8
**Confidence:** 3

**Summary:**

This paper addresses the core bottlenecks of physical consistency and parameter controllability in state-of-the-art text-to-video (T2V) generation. Existing models rely on appearance-level motion memorization without understanding underlying physical laws, leading to unrealistic motions (e.g., upward-falling objects) and poor generalization. To solve this, the authors propose NewtonGen, a hybrid framework integrating data-driven T2V generation with physics-informed Neural Newtonian Dynamics (NND). NND leverages unified neural ordinary differential equations (ODEs) combined with a residual MLP to learn latent dynamics from small-scale "physics-clean" data (generated via a custom simulator). It models 9-dimensional latent physical states (position, velocity, rotation, size, etc.) to predict physics-consistent trajectories. During inference, NND’s state predictions and scene prompts guide a motion-controlled T2V model (Go-with-the-Flow) to generate videos.

**Strengths:**

1. Directly addresses two critical limitations of state-of-the-art text-to-video (T2V) models—physical inconsistency (e.g., upward-falling objects, abrupt velocity changes) and lack of parameter controllability—which scaling alone cannot resolve. This fills a key research gap, as current models only learn motion distributions from visual appearance rather than underlying physical laws.

2. NND’s design (physics-informed linear neural ODEs + residual MLP) unifies diverse Newtonian motions (12 types tested, including uniform velocity, parabolic motion, rotation, deformation, and 3D motion) into a single framework. It efficiently learns latent dynamics from a small amount of "physics-clean" data, avoiding over-reliance on large-scale datasets.

3. Compares against 5 representative baselines (closed-source: Sora, Veo3; open-source: CogVideoX-5B, Wan2.2; physics-based: PhysT2V) across 12 motion types, ensuring fairness by standardizing generation settings.

4. Separates physical dynamics reasoning (NND, for state prediction) from video generation (motion-controlled model like Go-with-the-Flow). This design retains the visual quality of existing T2V models while injecting physical constraints and enabling easy integration with other T2V frameworks.

**Weaknesses:**

1. Relies on continuous ODEs and thus cannot handle discrete physical events (e.g., collisions, rebounds, explosions). This restricts its applicability to real-world scenarios involving interactive or abrupt physical interactions.
2.  NND is trained exclusively on data from a custom Python physics simulator (designed to avoid motion blur, noise, or background distractions). Performance on real-world videos (with clutter, occlusions, or motion blur) is unproven, as no experiments on real datasets are reported.
3. While NewtonGen outperforms baselines on physical consistency (PIS), it does not quantitatively evaluate visual quality (e.g., texture realism, shadow dynamics, background coherence) against competitors. This makes it unclear if physical consistency comes at the cost of visual fidelity.

**Questions:**

Please follow weakness.

---

> ### Author Response · Authors · 2025-11-21
>
> We sincerely thank reviewer **UcEn** for the thoughtful and constructive feedback. We will revise the paper accordingly and respond to the main points below.
>
> ---
>
> ### **W1. Cannot handle discrete physical events**
>
> As mentioned in the paper, NewtonGen is built on **continuous ODE-based dynamics**. Handling discrete physical events or complex multi-body interactions is highly challenging and remains an open research direction. One potential solution is to incorporate event-based ODEs or multi-body energy constraints, but this requires substantial new development. We acknowledge these limitations and view them as important future work for the community, rather than something a single model can currently solve.
>
> ---
>
> ### **W2. Experiments on real datasets**
>
> Thank you for the suggestion. We have added the results of training NND on **real-world videos**, please check L517 the revised paper. We also provide more real-world videos and results in the folder **Real\_World\_Test** in our Supplementary Material.
>
> Specifically, we used the real fall videos from *"PISA Experiments: Exploring Physics Post-Training for Video Diffusion Models by Watching Stuff Drop” (ICML 2025)*. Using these videos, we trained NND to model fall dynamics. As shown in **Fig. 5**, NND can effectively learn from real-world data and generalize to generate physically consistent falling motions in additional scenes.
>
> Real-world videos often contain motion blur, lighting variations, and other challenges. Consequently, the extracted ground-truth physical quantities (e.g., shape, area, position) are not perfectly accurate, which leads to slightly lower performance compared to simulation-only training. This also motivates our use of simulation data, which provides clean and precise physical supervision.
>
> In summary, training on real dynamic scenes to train our NewtonGen (NND) is feasible, but it is time-consuming and requires significant care (e.g., setting up scenes, measuring physical scales, capturing at high frame rates to avoid motion blur). Simulation provides a faster and more convenient way to obtain high-quality physical data.
>
> ---
>
> ### **W3. Evaluating visual quality**
>
> Thank you for pointing this out. In principle, our two-stage inference procedure does not change the visual rendering ability of the base model (Go-with-the-Flow). Following your suggestion, we added **object consistency** and **background consistency** metrics from VBench to Table 1. The results show that our method also brings clear improvements in visual fidelity.
>
> ---
>
> We will incorporate these clarifications and refinements in the revised version. Thank you again for your valuable feedback, and have a great day!

---

> > ### Comment · Reviewer_UcEn · 2025-11-27
> > **Feedback from reviewer**
> >
> > I have read the feedback, and I think my concerns have been addressed.

---

> > > ### Author Response · Authors · 2025-11-27
> > >
> > > Thank you for the reply, and have a great day!

---

### Official Review · Reviewer_r8si · 2025-10-31

**Soundness:** 3
**Presentation:** 3
**Contribution:** 4
**Rating:** 6
**Confidence:** 3

**Summary:**

This paper presents NewtonGen, a framework aimed at solving the lack of physical consistency and controllability in text-to-video (T2V) generation. The authors propose a framework that integrates a data-driven T2V model with learnable physical principles. Its core is the Neural Newtonian Dynamics (NND) module, a physics-informed neural ODE that models a 9-dimensional latent physical state (e.g., position, velocity, rotation, size). The NND is first trained on "physics-clean" synthetic data to learn these dynamics. During inference, this trained NND predicts a sequence of physical states from user-defined initial conditions, which are then converted to optical flow to guide a motion-controlled T2V generator. Experiments across 12 motion types demonstrate quantitative improvements in physical consistency and precise parameter control. The reviewer regards NewtonGen as a novel and well-motivated two-stage framework for injecting physical realism and controllability into video generation. The core contribution, the NND module, is noted as being cleverly designed, as it combines linear ODEs with a residual MLP to learn complex dynamics from synthetic data efficiently. The reviewer acknowledges that the framework shows significant quantitative improvements on the Physical Invariance Score (PIS) metric and possesses excellent qualitative controllability. However, the reviewer raises concerns about the framework's reliance on synthetic "physics-clean" data, which poses significant questions about the sim-to-real gap. Furthermore, the reviewer points out that the method is explicitly limited to continuous dynamics, excluding collisions or discrete interactions, which narrows the scope of its "Newtonian" claims

**Strengths:**

1. The NND module is cleverly designed, combining physics-informed linear neural ODEs with a residual MLP. This allows it to flexibly learn a wide range of dynamical systems, from simple to complex, with high data efficiency.

2. The framework achieves exceptional physical consistency across multiple motion types. In quantitative evaluations using the "Physical Invariance Score" (PIS) metric, NewtonGen's results significantly outperform all SOTA baselines and are very close to the simulated ground truth.

3. The model achieves precise control over physical parameters, addressing a key weakness in existing T2V models. Experiments demonstrate it can generate motions that accurately correspond to user-specified initial physical states (e.g., position, velocity, size).

**Weaknesses:**

1. Sim-to-Real Gap: The NND is trained exclusively on synthetic "physics-clean" data. The paper does not demonstrate or discuss how the framework would handle real-world, noisy videos, suggesting its current application may be limited to generation from scratch rather than editing or predicting existing real-world videos.

2. Limited Scope of Dynamics: The framework explicitly excludes discrete events such as collisions, rebounds, or multi-object interactions.

3. Approximation of 3D Motion: The paper claims to handle 3D motion, but the method is "equivalently realized through the combination of position and size control". This appears to be a 2.5D approximation (i.e., objects scaling larger as they move closer) rather than a true 3D representation capable of handling complex perspective rotations.

4. Misaligned Baseline Comparison: The paper compares NND against general-purpose T2V models like Sora and Veo3. These baselines were not optimized for fine-grained physical control, making the comparison somewhat misaligned.

**Questions:**

1. The authors should explicitly state in the main paper that the framework's current application is generation from parameters, not prediction from real video. The "continuous dynamics only" limitation should also be candidly discussed in the main text, with potential extensions for handling discrete events like collisions.

2. Add a "Perfect Oracle" Baseline: It is suggested to add a "perfect simulator" baseline to Table 1, where the ground-truth simulator optical flow is fed directly into the generator. This would quantify the performance gap between the learned NND and a "perfect" physics oracle.

3. The authors should report standard deviations for the PIS scores in Table 1, rather than only the median, to provide a clearer measure of generation stability.

---

> ### Author Response · Authors · 2025-11-21
>
> We sincerely thank reviewer **r8si** for the thoughtful and constructive feedback. We will revise the paper accordingly and respond to the main points below.
>
> ---
>
>
> ### **W1. Sim-to-Real Gap**
>
> Thank you for the suggestion. We have added experiments trained and tested on real-world videos, please check the **Real-world video training** part from L517 in the revised paper. We also provide more real-world videos and results in the folder **Real\_World\_Test** in our Supplementary Material.
>
> Specifically, we used the real-world falling videos from *"PISA Experiments: Exploring Physics Post-Training for Video Diffusion Models by Watching Stuff Drop” (ICML 2025)*. Using these videos, we trained NND to learn the dynamics. As shown in **Fig. 5**, with this real-world data, NND can effectively learn the dynamics and generalize to generate physically correct falling motions in additional scenes.
>
> We also show that real-world videos inevitably contain motion blur, lighting changes, and other challenges (please check the real-world videos' masks in Fig. 5). As a result, some ground-truth physical quantities (e.g., shape, area, position) extracted from these videos are not perfectly accurate, which leads to lower performance compared to our simulation-based training. This is also one of the motivations for using simulation data, which provides clean, precise physical knowledge.
>
> In summary, collecting real dynamic scenes to train NewtonGen (NND) is feasible, but it is time-consuming and requires care (e.g., setting up scenes, measuring physical scales, and capturing with high frame rates to avoid motion blur). Our simulation offers a faster and cheaper way to obtain training data.
>
> ---
>
> ### **W2. Limited Scope of Dynamics**
>
> As mentioned in the paper, NewtonGen is built on **continuous ODE-based dynamics**. Handling more discrete events or complex multi-body interactions is extremely challenging and is part of a broader research direction. We acknowledge these limitations and view them as future work for the community rather than something a single model can currently solve.
>
> ---
>
> ### **W3. Approximation of 3D Motion**
>
> We agree with the reviewer that our usage of “3D motion” is more accurately an **approximation**. One core reason we do not directly include **metric depth** is the lack of a robust video-based depth estimator that can reliably provide consistent foreground depth across frames. We will clarify this in the paper and update the terminology to “3D motion effect / approximation.” (please check L208)
>
> ---
>
> ### **W4. Misaligned Baseline Comparison**
>
> We understand the concern and would like to clarify. The PIS score in Table 1 **does not measure control**, and it simply checks whether certain physical quantities remain consistent in the generated video. It does not depend on the magnitude of the physical parameters. Therefore, this quantitative comparison is still fair. In **Fig. 4**, models like Sora do *not* provide physical parameter control. This figure actually strengthens the motivation and advantage of NewtonGen (Parameter Controllability) rather than creating an unfair comparison.
>
> While most academic papers cannot compete directly with closed-source state-of-the-art video models, to convince the reviewers and readers, we still include Veo3 and Sora to provide a more comprehensive and reliable comparison.
>
> ---
>
>
> ### **Q1. Clarifying the assumptions and limitations**
>
> Thank you for the reminder. We will explicitly state NewtonGen’s intended scope and limitations in the main paper. (please check L229, L537)
>
> ---
>
> ### **Q2. “Perfect Oracle” Baseline**
>
> As shown in Appendix Figures 7–18, the prediction errors of NND relative to ground-truth are very small. Its predicted dynamics can effectively serve as **a ground-truth-like simulator**, which aligns with your idea of a “Perfect Oracle.”
>
> In addition, Table 1 already includes a **reference score** computed on perfect simulated videos, which serves as an upper-bound reference for the PIS.
>
> ---
>
> ### **Q3. Standard deviations**
>
> Thank you for this suggestion. We have added all standard deviations to Table 1.
>
> ---
>
> We will incorporate these clarifications and refinements in the revised version. Thank you again.

---

### Official Review · Reviewer_tuJM · 2025-10-31

**Soundness:** 2
**Presentation:** 3
**Contribution:** 3
**Rating:** 6
**Confidence:** 3

**Summary:**

The paper presents a hybrid generative framework that combines physical dynamics with data driven video generation. Its main contribution is Neural Newtonian Dynamics (NND), a learnable ODE based module that predicts latent physical states and enforces physical constraints during text to video synthesis. This method seeks to close the gap between visual realism and physical consistency in generative video models.

**Strengths:**

1. The conceptual direction of embedding explicit dynamical models into video generation is strong and timely and the overall structure of the framework is clean and modular, separating physical dynamics reasoning from visual synthesis.

2. The experimental evaluation is broad and convincing for their claims, with meaningful metrics and visualizations (even though generated videos are slightly on the short side). Comparisons with “Physical simulation then Generation” methods would add context, the current analysis supports the claims.

**Weaknesses:**

1. The physical state encoder relies heavily on SAM2 and OpenCV to detect and extract geometric and kinematic properties from frames. While suitable for simple rigid-body scenarios, it is unclear how this mechanism could extend to richer physical properties such as elasticity, reflection, lighting, wave propagation, breaking, or liquid dynamics, leaving aside the potential extension of NND to multi-object scenarios. The current setup appears limited to tracking motion and deformation rather than modeling material or environmental physics, which constrains the framework’s generalization to more complex phenomena.

2. The mapping from physical prompts to initial physical states $Z_0$ is not well-detailed. It remains unclear how the system estimates or derives physical parameters like initial position, velocity, or angular speed from the prompt, and how sensitive the final generation is to inaccuracies in these estimates.

3. As indicated in Section 2, "Physical Simulation then Generation", "... generative models themselves lack any inherent physical reasoing or simulation capability: users must predefine the physical simulation parameters and rules for each scenario, and these setings cannot readily generalize to other context or different physical laws". I also fail to see how NND generalizes in this case, which I believe is also addmited in Appendix F, Question 4.

**Questions:**

1. A direct comparison with a traditional physical simulation following the same pipeline would be valuable. Currently, it is unclear where the NND demonstrates a distinct advantage, is it in computational efficiency, scalability to data (which physics simulations don't have), or improved fidelity compared to directly employing a physics engine? Considering that the NND itself is initially trained on physical simulations, clarifying where it provides genuine value beyond simulation would help better position the contribution.

---

> ### Author Response · Authors · 2025-11-21
>
> We sincerely thank reviewer **tuJM** for the thoughtful and constructive feedback. We will revise the paper accordingly and address the main points below.
>
> ---
>
> ### **W1. Generalization to more complex physical phenomena**
>
> NewtonGen is primarily designed for generating motions that follow **classical mechanics**, and we intentionally focus on salient, intuitive foreground motion and deformation. As you correctly pointed out, our physical state extraction currently relies on basic vision models/tools such as SAM2 and OpenCV to obtain properties like position, size, shape, and speed. At this stage, there is no unified vision model that can reliably extract richer physical attributes such as elasticity, reflectance, lighting behavior, wave propagation, breaking, or fluid dynamics. This limitation affects not only our work, but the broader area of physics-aware generation and understanding. In this paper, we therefore focus more on **foreground object dynamics** rather than complex environmental effects and background physics.
>
> Within this scope, NND learns from a relatively small amount of physics-clean data and provides a compact latent representation of foreground dynamics. When these latents are used to control the motion generator, the system can already produce fairly complex motions. We appreciate this concern and will more clearly state the goals and applicability of NewtonGen in the paper. Please see the revised *Problem Definition* at the beginning of the *Methodology* section (L229).
>
> ---
>
> ### **W2. Mapping prompts to initial physical states**
>
> During inference, the initial physical states are specified directly by the user through the prompt, or conceptually through an interactive canvas where a user could place the object in the frame, set its size, and draw velocity arrows. In other words, our current setup does not require estimating initial states from an input image or video, since our focus is **text-to-video generation**. We have clarified this point in L236.
>
> ---
>
> ### **W3. How NND generalizes**
>
> Our discussion of “Physical Simulation then Generation” is meant to highlight that such pipelines usually require manual configuration of the simulator for each scenario. Even for simple uniform motion (e.g., a boat moving on water versus a ball sliding on a table, as in Fig. 1(b)), the user typically has to set different physical solvers and parameters like mass, size, environmental forces, friction or drag, and other parameters separately for each case.
>
> In NewtonGen, however, these scenarios can share a single set of neural dynamics, which allows a certain level of cross-scene generalization and consistent motion behavior. Moreover, different types of dynamics such as free fall and friction-induced deceleration would usually require different simulator configurations in the previous methods, whereas NND offers a **single neural dynamical formulation** that can cover multiple types of dynamics within one unified framework.
>
> Appendix F, Q&A 4 discusses a different limitation, namely that the current version of NewtonGen cannot yet handle discontinuous dynamics such as hard collisions. This is orthogonal to the generalization behavior described above.
>
> ---
>
> ### **Q1. Comparison with traditional physical simulation**
>
> To the best of our knowledge, there is currently no established baseline for physically consistent and controllable text-to-video generation. Conceptually, NND’s predictions can be viewed as an accurate learning-based physical simulator.
>
> Following your suggestion, we summarize where NND can offer advantages compared to a traditional physical simulation-based approaches:
>
> 1. **Unified dynamics.**
>    The ODE+MLP design naturally accommodates multiple linear and nonlinear dynamics without switching simulators or hand-tuned rules.
>
> 2. **Computational efficiency and compact representation.**
>    NND models the dominant scene dynamics using only a 9-dimensional latent state, without operating on textures or background geometry, and enables real-time inference.
>
> 3. **Scalability to data.**
>    Classical physical simulation relies on idealized, handcrafted physical rules, whereas NND can learn dynamics from simulation or real-world videos (we also add more results trained on real-world videos in the revised version paper), allowing it to capture complex or hard-to-model effects (e.g., nonlinearities, perturbations).
>
> These advantages are also reflected and briefly discussed in Section “Preliminary Concepts” in the paper.
>
> ---
>
> We will incorporate these clarifications and refinements into the revision. Thank you again for your detailed and thoughtful review — we truly appreciate your time and feedback.

---

### Meta-Review · Area_Chair_pwTc · 2025-12-24

**Summary:**

**Summary**: The paper presents NewtonGen, a novel framework for producing physically consistent and controllable videos upon the existing state-of-the-art text-to-video (T2V) generators. The key innovation is to build a physics-informed Neural Newtonian Dynamics (NND), which combines the ordinary differential equations (ODES) with a residual MLP to learn the dynamic physical states from the given text prompt. The method achieves better quantitative results on PISA benchmark and shows more visual results for the in-the-wild videos.

**Main strengths**:
- To improve the physical consistency and parameter controllability, the explicit modeling of Newtonian motions is important and interesting.
- A simple NND architecture, combined with ODES and MLP, unifies diverse motion types into a single network, and then predicts the corresponding physically plausible optical flow.
- Comprehensive empirical validation across different state-of-the-art video generators.

**Main weaknesses**:
- Sim-to-Real Gap: The NND architecture is trained on synthetic data, which may not generalize well to real video.
- Simplified motion: The current experiments are mainly on rigid and single objects, which cannot handle complex motions.

**Suggested decision**: This paper received initial scores of 6 (tuJM), 6 (r8si), 8 (UcEn), and 2 (CpA9). While reviewer CpA9 claimed "the approach is simple and inherently unscalable", I view this paper as an initial step toward explicitly modeling Newton's law in video generation. Real-world motion governed by physical laws is inherently complex, especially when involving multiple objects; starting with simplified settings is a reasonable and necessary strategy. Hence, I recommend the final decision as "accept".

**Reviewer Concerns:**

- **Sim-to-Real gap (tuJM, r8si, UcEn)**: Not fully addressed, but added more real video examples.

- **Misaligned comparison (r8si)**: Addressed.

- **Visual quality (UcEn)**: Addressed.

- **Simplified motion (tuJM, r8si, UcEn, CpA9)**: This is the main concern posed by all reviewers, which is still the big limitation of this work, as it only focuses on a single rigid object with simple motion.

- **not much novelty (CpA9)**: Even considering the paper as an "optical flow conditioning" paper, how to generate the physically plausible optical flow from the text is still meaningful.

**Reviewer Scores:**

The paper initially received scores of 6 (tuJM), 6 (r8si), 8 (UcEn), and 2 (CpA9).
Reviewer UcEn calimed that all concerns have been addressed in the rebuttal.
In contrast, Reviewer CpA9 maintains the view that the approach is “overly simplistic and inherently unscalable,” and argues that, if interpreted as "an optical-flow conditioning paper, there is not much novelty in it".
As a result, these reviewers are likely to retain their original scores.

---

### Decision · Program_Chairs · 2026-01-26

Accept (Poster)